# ReSeek: A Self-Correcting Framework for Search Agents with Instructive Rewards

## Abstract

Search agents powered by Large Language Models (LLMs) have demonstrated significant potential in tackling knowledge-intensive tasks. Reinforcement learning (RL) has emerged as a powerful paradigm for training these agents to perform complex, multi-step reasoning. However, prior RL-based methods often rely on sparse or rule-based rewards, which can lead agents to commit to suboptimal or erroneous reasoning paths without the ability to recover. To address these limitations, we propose **ReSeek**, a novel self-correcting framework for training search agents. Our framework introduces a self-correction mechanism that empowers the agent to dynamically identify and recover from erroneous search paths during an episode. By invoking a special **JUDGE** action, the agent can judge the information and re-plan its search strategy. To guide this process, we design a dense, instructive process reward function, which decomposes into a correctness reward for retrieving factual information and a utility reward for finding information genuinely useful for the query. Furthermore, to mitigate the risk of data contamination in existing datasets, we introduce **FictionalHot**, a new and challenging benchmark with recently curated questions requiring complex reasoning. Being intuitively reasonable and practically simple, extensive experiments show that agents trained with ReSeek significantly outperform SOTA baselines in task success rate and path faithfulness. Our code and dataset are available at https://anonymous.4open.science/r/Re-Search-5A0F.

## 1 Introduction

Large Language Models (LLMs) (Brown et al., 2020; OpenAI, 2022; Ouyang et al., 2022; Guo et al., 2025; Yang et al., 2025) have demonstrated unprecedented capabilities in natural language understanding and generation, yet they are inherently limited by their static, pre-trained knowledge, which can be outdated or lead to factual hallucinations (Borgeaud et al., 2022; Zhao et al., 2024; Maleki et al., 2024; Zhang et al., 2025b). Search-augmented agents, which empower LLMs to interact with external tools like search engines, have emerged as a powerful paradigm to overcome these limitations (Li et al., 2025c; Zheng et al., 2025; Jin et al., 2025; Luo et al., 2025). By dynamically retrieving and reasoning over up-to-date information, these agents can tackle complex, knowledge-intensive tasks that are beyond the reach of standalone LLMs.

Despite their promise, the prevailing methods often falter in scenarios requiring complex reasoning. Early approach, Retrieval-Augmented Generation (RAG) enhance LLMs with external knowledge but are often limited to a single retrieve-then-generate cycle, lacking the procedural capability for sequential decision-making or error correction. More advanced agents trained with Reinforcement Learning (RL) attempt to address this by learning a policy over a sequence of actions. However, these agents often suffer from the limitations of their reward structures. Relying on rule-based rewards (e.g., a final answer's correctness) or simple, heuristic-based process rewards provides insufficient guidance for intermediate steps. Consequently, if an agent makes an early mistake—such as pursuing a misleading search query—it tends to irrevocably commit to this erroneous path, leading to a cascade of errors and an ultimately incorrect answer. This inability to self-assess and recover from mistakes is a fundamental bottleneck hindering their reliability and performance. As shown in Figure 1 (a) and (b), both RAG and Search-R1(Jin et al., 2025) query for "creator of Saddle Rash." The retrieved documents center almost exclusively on describing the show itself and contain no information about the director's birth date, ultimately leading to an unsuccessful response.

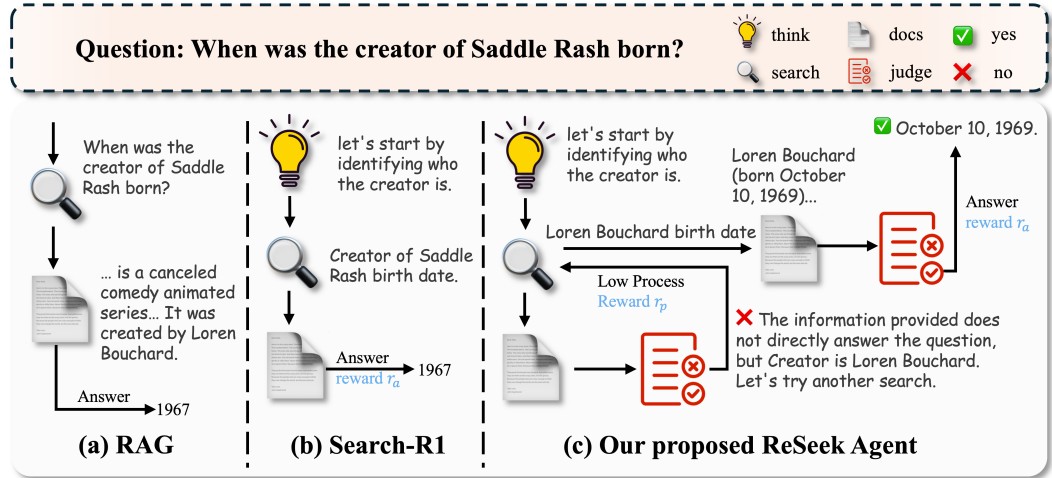

Figure 1: **A comparison of reasoning processes on a multi-hop question about an obscure entity.** Standard RAG (a) fails as it cannot perform sequential reasoning. Vanilla agent like Search-R1 (b) reasons sequentially but gets stuck on its initial path. In contrast, our agent (c) demonstrates robust self-correction: it uses a low process reward $(r_p)$ to identify the unproductive intermediate step, triggers a JUDGE action to revise its strategy, and successfully navigates to the correct answer. The full trace for this example is provided in Appendix A.5.2.

To address these limitations, we propose ReSeek, a novel self-correcting framework for training search agents. The core of ReSeek is a dynamic self-correction mechanism, centered around a special JUDGE action. This action empowers the agent to pause, evaluate the utility of its most recent finding, and dynamically adapt its strategy if the path is unproductive or incomplete. As shown in Figure 1 (c), the initial search failed to yield a direct answer but revealed the creator's name: Loren Bouchard. This prompted a JUDGE action. The agent evaluated this new information as useful but insufficient. Consequently, it adapted its strategy by formulating a new query, "Loren Bouchard birth date," and ultimately retrieved the correct answer. This entire process is guided by a **dense and instructive reward function**. The reward is multi-faceted, decomposing into two key components: a correctness reward that encourages factual retrieval, and a utility reward that incentivizes finding query-relevant information. This dual-component structure provides step-by-step feedback, teaching the agent to find not just correct facts, but the right facts at the right time.

Another challenge lies in the evaluation of these advanced agents. Many existing datasets for knowledge-intensive tasks are at risk of data contamination, as their contents may have been included in the training corpora of the very LLMs we seek to evaluate. This creates a scenario where high performance might reflect memorization rather than genuine reasoning ability. To address this evaluation challenge, we first introduce **FictionalHot**, a new benchmark composed of recently curated questions about fictional entities that require complex, multi-hop reasoning. Its design inherently mitigates the risk of data contamination, forcing agents to rely solely on their procedural search and reasoning capabilities. We then integrate FictionalHot with several established public datasets to construct a comprehensive benchmark. This combined suite allows us to assess agent performance across a wide spectrum of tasks while ensuring a rigorous test of genuine reasoning ability, free from memorization artifacts. Our contributions are summarized as follows:

- We propose **ReSeek**, a novel framework that equips search agents with a self-correction mechanism, enabling them to dynamically adapt their search strategy and recover from unproductive reasoning paths.
- We design a **dense, process-based reward function** that guides the agent by providing distinct feedback on the factual correctness and contextual utility of retrieved information.
- We introduce **FictionalHot**, a new benchmark designed to facilitate a fair and challenging evaluation of search agents' reasoning abilities by mitigating the risk of data contamination.
- Through extensive experiments, we demonstrate that agents trained with ReSeek outperform SOTA baselines in task success rate and the faithfulness of their reasoning paths.

## 2 RELATED WORK

### 2.1 RETRIEVAL-AUGMENTED GENERATION AND SEARCH AGENTS

To mitigate the inherent knowledge limitations and factual hallucinations of Large Language Models (LLMs) (Zhang et al., 2023; Ji et al., 2023; Bang et al., 2023), a significant body of research has focused on integrating external knowledge. Pioneering Retrieval-Augmented Generation (RAG) frameworks (Lewis et al., 2020; Guu et al., 2020; Karpukhin et al., 2020) enhance LLM responses by retrieving relevant documents. However, their typical single-step "retrieve-then-generate" cycle limits their efficacy on tasks requiring complex, multi-step reasoning (Jiang et al., 2023; Asai et al., 2023). To address this limitation, advanced Search Agents have been developed (Shinn et al., 2023), which decompose complex tasks by interleaving search and reasoning steps (Yao et al., 2022). Works such as WebThinker (Li et al., 2025c), DeepResearcher (Zheng et al., 2025; Jin et al., 2025), Search-o1 (Li et al., 2025b), and ZeroSearch (Sun et al., 2025) significantly improve performance on knowledge-intensive tasks by empowering LLMs to plan and execute multi-step strategies. Despite enhancing the dynamism of the reasoning path, these agents predominantly follow a simple execution flow. This rigidity means an early misstep, such as pursuing a flawed search query, can lead to a cascade of errors, as they lack a mechanism for intra-episode assessment and correction.

### 2.2 SEARCH WITH REINFORCEMENT LEARNING

Reinforcement Learning (RL) has emerged as a powerful paradigm for training search agents capable of sequential decision-making. By modeling the search process as a Markov Decision Process, RL can optimize an agent's policy to maximize cumulative rewards. Recent works such as Search-R1 (Jin et al., 2025), ToolRL (Qian et al., 2025), and others (Mai et al., 2025; Zeng et al., 2023) have successfully applied RL to train LLMs to use search tools. A key challenge in this paradigm, however, lies in the design of the reward function. Many existing methods rely on sparse, outcome-based rewards (*e.g.*, correctness of the answer) (Jin et al., 2025; Mai et al., 2025). While straightforward to implement, such rewards provide poor credit assignment for intermediate steps, offering little guidance for navigating complex reasoning paths (Ouyang et al., 2022). Consequently, research has shifted towards denser guidance through process supervision and self-correction mechanisms (Chen et al., 2023). For instance, Backtracking Correction (Feng et al., 2025) refines the reasoning chain in a post-hoc manner, optimizing backwards from a completed trajectory. In contrast, our proposed Re-Search framework enables dynamic, intra-episode self-correction via a JUDGE action.

## 3 METHODOLOGY

In this section, we detail our proposed framework for training Large Language Model (LLM) agents to perform complex, multi-step tasks. Our approach is centered on enhancing the agent's reasoning and decision-making capabilities through a novel reinforcement learning paradigm. We begin by formally defining the problem.

### 3.1 PROBLEM FORMULATION

To enhance the agent's complex problem-solving abilities, we employ a policy optimization framework inspired by methods like Generative Representational Policy Optimization (GRPO). In this setup, the agent is represented by a policy $\pi_\theta$, parameterized by a Large Language Model. The policy's role is to generate an action $a_t$ at each step $t$ based on the current state $s_t$.

A key contribution of our framework is the introduction of a special JUDGE action, which empowers the agent to pause, evaluate the gathered information, and dynamically re-plan its subsequent steps. To train the agent to leverage this capability, we optimize its policy $\pi_\theta$ against a reference policy $\pi_{\text{ref}}$ using the following objective:

$$\max_\theta \mathbb{E} x \sim \mathcal{D}, y \sim \pi\theta(\cdot|x)[R(x,y)] - \beta D_{KL}[\pi_\theta(y|x)||\pi_{\text{ref}}(y|x)] \tag{1}$$

In this objective, the policy $\pi_\theta$ generates a trajectory $y$ for a given problem xxx from the dataset $\mathcal{D}$. The term $R(x,y)$ represents the cumulative reward, which is the sum of step-wise rewards designed

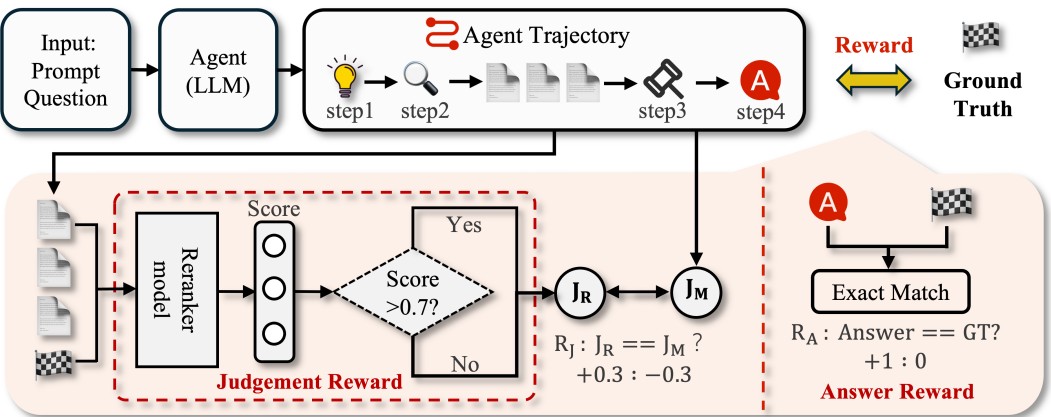

Figure 2: **Training the agent's self-evaluation capability.** We train the agent via policy optimization to master the JUDGE action. A reward signal is generated by comparing the agent's judgment against an "ideal" one, which is determined by the rerank score between the current search observation and the GT answer. This reward guides the policy to learn effective self-correction.

to provide dense signals for both tool use and the strategic invocation of the JUDGE action. The Kullback-Leibler (KL) divergence term, controlled by the hyperparameter $\beta$, regularizes the policy update against the reference policy $\pi_{\text{ref}}$ to ensure training stability.

## 3.2 JUDGE ACTION FOR SELF-CORRECTION

A core challenge for autonomous agents is ensuring the reliability and efficiency of their reasoning process. Even a single suboptimal action, driven by hallucination or misinterpretation, can derail an entire task, regardless of its overall complexity. To operate robustly, it is therefore crucial for an agent to possess a degree of meta-cognitive awareness—the ability to assess the utility of its intermediate steps and dynamically adjust its strategy. Our work addresses this by instilling a foundational layer of self-correction, enabling a more reflective and adaptive reasoning process that enhances performance across a spectrum of task difficulties.

To achieve this, we introduce the JUDGE action, a mechanism designed to serve as the agent's instrument for introspection. The JUDGE action converts the agent's reasoning from a static, linear chain into a *dynamic self-correction loop*. This mechanism does not rely on complex state backtracking, but rather on a simple yet powerful principle: *selective attention to history*. The agent learns to disregard uninformative steps when formulating its next action. We formalize this by having the policy condition its next action $a_{t+1}$ on a *dynamically assembled* context $\mathcal{C}_t$:

$$a_{t+1} \sim \pi(\cdot|\mathcal{C}_t) \quad \text{where} \quad \mathcal{C}_t = \tau_{t-1} \oplus \mathbb{I}(j_t \neq \text{'bad'}) \cdot o_t \tag{2}$$

In this formulation, the context $\mathcal{C}_t$ is assembled on-the-fly that $\mathbb{I}(\cdot)$ is the indicator function and $\oplus$ denotes concatenation. A favorable judgment appends $o_t$ and enriches the evidence available to the policy. An unfavorable judgment omits it, so the next action $a_{t+1}$ is conditioned on $\tau_{t-1}$ alone. Repeating this at every step yields a lightweight self-correction loop in which informative observations enter the context and unhelpful ones are filtered out.

## 3.3 REWARD FUNCTION FOR SELF-CORRECTION

For the JUDGE action to be effective, the agent's policy must learn to produce judgments that are aligned with actual task progress. Relying solely on the final task outcome provides a sparse and often delayed signal, making it difficult to learn the nuanced skill of step-by-step self-assessment. To address this, we introduce a dense, intermediate reward signal specifically designed to cultivate the agent's self-correction faculty. The core idea is to reward the agent for making judgments that match an objective, post-hoc evaluation of its actions.

We formalize this with a self-correction reward function, $R_{judge}$, which is given whenever a JUDGE action is performed:

$$R_{judge}(j_t, j_t^*) = \begin{cases} +R_{match} & \text{if } j_t = j_t^* \\ -R_{mismatch} & \text{if } j_t \neq j_t^* \end{cases} \tag{3}$$

This function provides a positive reward when the agent's judgment $j_t$ matches an objective Ideal Judgment $j_t^*$, and a negative reward otherwise. To establish this ideal judgment, we first compute a utility score, $s_t = \texttt{rerank\_score}(o_t, \text{GT})$, which quantifies the objective value of the observation (information) $o_t$ by measuring its semantic relevance to the ground-truth answer. This score is then mapped to a discrete label (good or bad) using a predefined threshold of 0.7.

## 3.4 STRUCTURED PROMPTING

The self-correction mechanism fundamentally relies on the agent's ability to generate structured trajectories of actions and observations. To this end, we designed the structured prompt in Table 1 to enforce a strict, self-corrective reasoning prompt. It achieves this by incorporating two key architectural constraints: first, a mandatory <judge> action creates an explicit self-assessment checkpoint after every information retrieval step. Second, strict conditional rules (Rule 4) make the agent's subsequent actions contingent on the judgment's outcome. The explicitness of these rules ensures immediate and reliable trajectory generation, even from an untrained LLM.

Table 1: **Self-corrective agent prompt.** After each search, the model performs a <judge> assessment and follows conditional rules to either continue searching or provide an answer. The Question placeholder is replaced at runtime with the current query.

---

Answer the given question step by step. Instructions:
1. First, conduct reasoning inside <think> and </think> tags whenever you receive new information.
2. If you need external knowledge, you can search using <search> query </search>.
3. When you receive search results, evaluate their usefulness and put your judgment inside <judge> Yes </judge> or <judge> No </judge> tags.
4. Based on your judgment, follow these strict rules:
   a. If the information is useful AND you now have sufficient information to provide a complete final answer, proceed directly to step 5.
   b. If the information is useful BUT you still need more details, you MUST search again with <search> ... </search>
   c. If the information is not useful, you MUST search again with <search> ... </search>. You MUST NOT provide an answer in this case.
5. Provide your final answer in <answer> ... </answer> tags. The <answer> tag marks the end of the task. After providing the <answer>, you MUST stop and generate no further text.
Question: [question]

---

## 3.5 FICTIONALHOT BENCHMARK

Robust evaluation of search agents is hampered by inconsistency in experimental settings. Table 2 collates representative setups, highlighting variation in (i) corpora—from static Wikipedia snapshots (e.g., 2018, 2019) to the non-reproducible, live Internet; (ii) test sets—either a broad Set A (NQ(Kwiatkowski et al., 2019), TriviaQA(Joshi et al., 2017), PopQA(Mallen et al.), HotpotQA(Yang et al.), 2Wiki(Ho et al.), Musique(Trivedi et al.), Bamboogle(Press et al.)) or a focused multi-hop Set B (HotpotQA, 2Wiki, Musique, Bamboogle); (iii) training regimes—ranging from no training to single or multi-dataset setups (e.g., HotpotQA, 2Wiki, NQ, TriviaQA); and (iv) metrics—spanning Exact Match and F1 to model-based judgments such as LLM-as-a-judge (LJ). While this diversity reflects rapid exploration, it hinders apples-to-apples comparison across papers.

Beyond standardization, a deeper challenge is data contamination, where high scores on existing benchmarks can reflect memorized pretraining knowledge rather than genuine procedural reasoning. To address both issues, we introduce FictionalHot, a closed-world benchmark that grounds all questions in a newly generated synthetic corpus, thereby (i) fixing a single, reproducible knowledge source and (ii) mitigating contamination by populating it with fictitious entities absent from pretraining. This design forces agents to rely on procedural reasoning over the provided documents and enables cleaner, apples-to-apples evaluation of search and answer capabilities.

Table 2: **Variety in Experimental Setups of Prior Works.** LJ means the metric of LLM-as-a-judge. This Table highlights the diversity in test sets, training sets, corpus, and evaluation metrics.

| Methods | Test Sets | Training Sets | Corpus | Metrics |
| --- | --- | --- | --- | --- |
| Search-o1 (Li et al., 2025b) | Set A | None | Internet | Exact Match |
| R1-Searcher (Song et al., a) | Set B | HotpotQA, 2wiki | 2019-wiki | Cover Exact Match, LJ |
| Search-R1 (Jin et al., 2025) | Set A | NQ, TriviaQA | 2018-wiki | Exact Match |
| ReSearch (Chen et al., 2025) | Set B | Musique | 2018-wiki | Exact Match, LJ |
| R1-Searcher++ (Song et al., b) | Set B | HotpotQA, 2wiki | 2019-wiki | F1, LJ |
| Deepresearcher (Zheng et al., 2025) | Set A | NQ, TriviaQA, HotpotQA, 2wiki | Internet | F1, MBE |
| ZeroSearch (Sun et al.) | Set A | NQ, TriviaQA | Internet | Substring Exact Match |

The construction of FictionalHot follows a three-step pipeline, as illustrated in the Figure 3. First, we draw a 10% random sample of seed questions from the seven benchmarks mentioned before. Next, these questions are paraphrased by GPT-5. This core step replaces real-world entities with plausible, fictional ones (e.g., 'Taylor Swift' becomes 'Lila Starling') while preserving the original question's reasoning structure. Crucially, GPT-5 also generates new Wikipedia-style documents for these fictional entities, creating a new, self-contained fact (e.g., setting the album release to '2007') that serves as the basis for the new golden answer. Finally, to create the closed-world corpus, these synthetic samples are inserted into the 2018 Wikipedia corpus. The complete Hot Benchmark is then formed by combining these Fictional Samples (FictionalHot) with the original seven benchmarks.

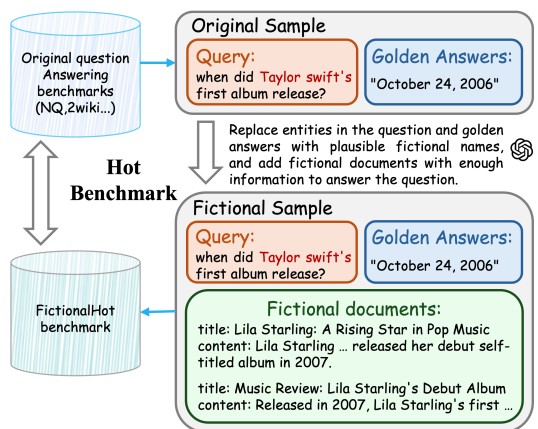

Figure 3: **The FictionalHot benchmark construction process:** transforming a real-world question answer sample into a fictional sample with fictional question and documents.

## 4 EXPERIMENTS

### 4.1 EXPERIMENTAL SETUP

**Evaluation Benchmarks.** We evaluate on two benchmark families: (i) the classic open-domain QA suites listed in 3.5: single-hop QA (NQ(Kwiatkowski et al., 2019), TriviaQA(Joshi et al., 2017), PopQA(Mallen et al.)) and multi-hop QA (HotpotQA(Yang et al.), 2Wiki(Ho et al.), Musique(Trivedi et al.), and Bamboogle(Press et al.)); and (ii) our FictionalHot benchmark, which mitigates contamination via a closed-world, synthetic corpus. To ensure fair comparison, we follow (Jin et al., 2025) and adopt Exact Match (EM) as the primary metric. A prediction is considered correct if its normalized string exactly matches any normalized reference answer. Normalization applies lowercasing, removes punctuation and articles, and collapses whitespace.

**Training and Evaluation Setup.** Followed (Jin et al., 2025), we fine-tune on a unified training set that merges the NQ and HotpotQA training splits. We conduct experiments using two types of models: Qwen-2.5-3B-Instruct and Qwen-2.5-7B-Instruct (Qwen et al., 2025). At test time, retrieval top-k is set to k=3 with a maximum of T=4 tool-use turns per question. Experiments run on 16 Ascend 910B NPUs; the search backend uses E5 embeddings(Wang et al., 2022), and the evaluation corpus is the 2018 Wikipedia corpus (wiki-18) (Karpukhin et al., 2020). Unless otherwise noted, GRPO is used as the default RL algorithm, and a detailed comparison with PPO is provided in Appendix A.2. Additional details are provided in Appendix A.1.

**Baselines.** We compare with four baseline families: Vanilla prompting (zero-shot direct answering and Chain-of-Thought with no external search) (Karpukhin et al., 2020); Single-pass RAG (retrieve once, then generate conditioned on the top-k passages) (Lewis et al., 2020); Agentic search (multi-step search–reason loops such as ReAct-style planners, without RL tuning) (Chung et al., 2024; Li et al., 2025b); and RL-tuned policies (Jin et al., 2025; Chen et al., 2025; Sun et al.).

Table 3: **Overall performance comparison of ReSeek against baselines across a comprehensive suite of QA benchmarks.** Results are reported in Exact Match (EM) for both Qwen2.5-7B and 3B backbones. The best results are in **bold**, and our method is highlighted with a bule row.

| Methods | General QA | | | Multi-Hop QA | | | | | |
|---|---|---|---|---|---|---|---|---|---|
| | NQ | TriviaQA | PopQA | HotpotQA | 2wiki | Musique | Bamboogle | FictionalHot | Avg. |
| **Qwen2.5-7b-Instruct** | | | | | | | | | |
| Direct Inference | 0.134 | 0.408 | 0.140 | 0.183 | 0.250 | 0.031 | 0.120 | 0.001 | 0.158 |
| CoT | 0.048 | 0.185 | 0.054 | 0.092 | 0.111 | 0.022 | 0.232 | 0.001 | 0.093 |
| RAG | 0.349 | 0.585 | 0.392 | 0.299 | 0.235 | 0.058 | 0.208 | 0.012 | 0.267 |
| SFT | 0.318 | 0.354 | 0.121 | 0.217 | 0.259 | 0.066 | 0.112 | 0.003 | 0.181 |
| R1 (Guo et al., 2025) | 0.270 | 0.537 | 0.199 | 0.237 | 0.292 | 0.072 | 0.293 | 0.003 | 0.238 |
| Search-o1 (Li et al., 2025b) | 0.151 | 0.443 | 0.131 | 0.187 | 0.176 | 0.058 | 0.296 | 0.020 | 0.183 |
| Search-R1 (Jin et al., 2025) | 0.393 | 0.610 | 0.397 | 0.370 | **0.414** | 0.146 | 0.368 | 0.034 | 0.342 |
| ZeroSearch (Sun et al.) | 0.436 | **0.652** | 0.488 | 0.346 | 0.352 | 0.184 | 0.278 | 0.031 | 0.346 |
| ReSeek | **0.469** | 0.640 | **0.501** | **0.389** | 0.382 | **0.185** | **0.392** | **0.061** | **0.377** |
| **Qwen2.5-3b-Instruct** | | | | | | | | | |
| Direct Inference | 0.106 | 0.288 | 0.108 | 0.149 | 0.244 | 0.020 | 0.024 | 0.001 | 0.118 |
| CoT | 0.023 | 0.032 | 0.005 | 0.021 | 0.021 | 0.002 | 0.000 | 0.001 | 0.013 |
| RAG | 0.348 | 0.544 | 0.387 | 0.255 | 0.226 | 0.047 | 0.080 | 0.008 | 0.237 |
| SFT | 0.249 | 0.292 | 0.104 | 0.186 | 0.248 | 0.044 | 0.112 | 0.001 | 0.155 |
| R1 (Guo et al., 2025) | 0.210 | 0.449 | 0.171 | 0.208 | 0.275 | 0.060 | 0.192 | 0.003 | 0.196 |
| Search-o1 (Li et al., 2025b) | 0.238 | 0.472 | 0.262 | 0.221 | 0.218 | 0.054 | 0.019 | 0.010 | 0.187 |
| Search-R1 (Jin et al., 2025) | 0.341 | 0.545 | 0.378 | 0.324 | 0.319 | 0.103 | 0.264 | 0.037 | 0.288 |
| ZeroSearch (Sun et al.) | 0.414 | **0.574** | **0.448** | 0.274 | 0.300 | 0.098 | 0.111 | 0.030 | 0.281 |
| ReSeek | **0.415** | 0.553 | 0.434 | **0.328** | 0.298 | **0.103** | **0.304** | **0.059** | **0.312** |

## 4.2 MAIN RESULTS

We evaluate ReSeek across eight open-domain QA benchmarks spanning single- and multi-hop settings, using Qwen2.5-7B-instruct and Qwen2.5-3B-instruct backbones.

**ReSeek achieves SOTA performance.** We evaluate ReSeek across eight open-domain QA benchmarks spanning single- and multi-hop settings. ReSeek attains the highest average accuracy across both backbones: 0.377 for 7B compared to 0.346 for ZeroSearch, and 0.312 for 3B compared to 0.281. It consistently excels on multi-hop benchmarks, notably HotpotQA and Bamboogle across both model scales, which highlights the benefits of our repeated-search with self-correction paradigm. On single-hop datasets ZeroSearch performs well, which aligns with its design focus.

**FictionalHot isolates reasoning ability from model scale and data leakage.** On FictionalHot, ReSeek scores 0.061 with 7B and 0.059 with 3B. This near-identical performance, unlike the large gaps seen on standard datasets, indicates that FictionalHot successfully isolates reasoning ability from the stored knowledge that typically correlates with model scale. In contrast, Direct Inference on TriviaQA reaches 0.408 for 7B and 0.288 for 3B, while scoring almost zero on FictionalHot ( 0.001). This pattern highlights likely training-data overlap in TriviaQA, whereas FictionalHot—constructed from synthetic, fictional entities with no possibility of leakage—provides a cleaner measure of retrieval and reasoning, underscoring the importance of our benchmark.

## 4.3 ABLATIVE ANALYSIS

**Ablation Study on the Reranker Component.** To assess the overall effectiveness of our reward function and the specific choice of its reranker component, we conduct an ablation study presented in Table 4. We compare our primary method, ReSeek (with BGE-Reranker), to three variants: (i) None, removing the reranker; (ii) Qwen-Reranker (Zhang et al., 2025a), a comparable neural reranker; and (iii) Regex-based, a heuristic-driven reward mechanism that does not use any neural reranker.

The Regex-based method works by parsing the agent's reasoning trace to identify pairs of retrieved information and the agent's judgment on its usefulness (i.e., "Yes" or "No"). It then determines the "ground-truth usefulness" of the information by checking for the literal presence of the final answer string within the retrieved content. The agent is rewarded for correct judgments (e.g., saying "Yes" to information containing the answer) and penalized for incorrect ones. Notably, this method

Table 4: **Ablation study on the reranker component of our reward function.** Our method, ReSeek, uses the BGE-Reranker. We compare it against variants with a different neural reranker (Qwen), a heuristic reranker (Regex-based), or no reranker at all (None).

| Methods | General QA | | | Multi-Hop QA | | | | | |
|---|---|---|---|---|---|---|---|---|---|
| | NQ | TriviaQA | PopQA | HotpotQA | 2Wiki | Musique | Bamboogle | FictionalHot | Avg. |
| None (w/o Reranker) | 0.391 | 0.495 | 0.362 | 0.255 | 0.218 | 0.081 | 0.243 | 0.025 | 0.259 |
| Regex-based | 0.410 | 0.541 | 0.422 | 0.320 | 0.291 | 0.093 | 0.288 | 0.042 | 0.301 |
| Qwen-Reranker | 0.413 | **0.557** | 0.432 | 0.326 | **0.301** | 0.101 | 0.302 | 0.057 | 0.311 |
| ReSeek (Ours, w/ BGE) | **0.415** | 0.553 | **0.434** | **0.328** | 0.298 | **0.103** | **0.304** | **0.059** | **0.312** |

applies asymmetric penalties, imposing a larger penalty for incorrectly judging useless information as useful (a false positive) than for failing to identify useful information (a false negative), thereby discouraging the agent from "hallucinating" evidence.

As shown in Table 4, the results reveal a clear performance hierarchy. The "Regex-based" method improves upon the "None" baseline, demonstrating the value of a simple lexical signal. However, the most significant gains are achieved by neural rerankers (BGE and Qwen). This highlights that semantic understanding, which goes beyond simple string matching, is critical for accurately assessing document relevance in complex queries.

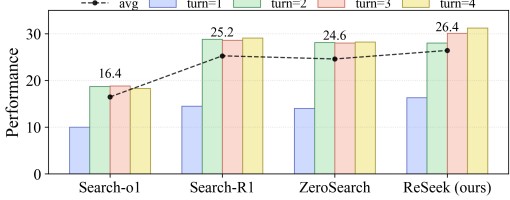

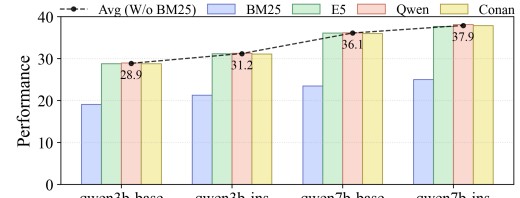

Figure 4: **Ablation study on the effect of the number of turns on model performance.** We evaluate multiple methods with turn budgets from 1 to 4 using qwen2.5-3b-instruct, reporting the average performance across all datasets.

Figure 5: **Ablation study on search-embedding choice and base/instruction models.** We evaluate our method on the Wiki18 corpus across different backbone and embedding models over all datasets. The dashed line denotes the mean performance (excluding BM25).

**Interaction Turns Study.** We perform an ablation over the number of turns to isolate the effect of the action budget and to test whether models can leverage iterative self-correction. Here, turns denotes the maximum number of actions the model may execute for a query. This setup tests whether extra action steps help the model recheck evidence and revise hypotheses, or whether performance already saturates with a minimal search-then-answer cycle. As shown in Figure 4, the baselines improve substantially from one to two turns, then show little or no gain at three and four turns, consistent with their two-step workflow (typically one turn to search and one to answer). In contrast, ReSeek improves monotonically from one through four turns, indicating stronger self-correction: with more turns it re-queries evidence when uncertain, refines its plan, and revises its answer. The average performance mirrors this trend, with ReSeek achieving the highest mean score, demonstrating that our method converts a larger turn budget into genuine gains rather than redundancy.

**Sensitivity to the retrieval encoder.** We ablate the search embedding on the Wiki18 corpus while holding the rest of the pipeline fixed and evaluating across qwen3b/7b base and instruction backbones. As shown in Figure 5, BM25 (Robertson et al., 2009) consistently underperforms the dense retrievers, reflecting lexical mismatch and limited semantic coverage. Among dense encoders, E5 (Wang et al., 2022), Qwen (Zhang et al., 2025a), and Conan (Li et al., 2025a) are close, with Qwen slightly ahead of E5. Because our datasets are entity-centric, retrieval isn't particularly hard and performance changes little with reasonably capable embeddings.

**Base vs. instruction-tuned backbones.** We focus on the difference between base and instruction-tuned backbones. As shown in Figure 5, averaging over dense embeddings (excluding BM25), instruction-tuned models consistently outperform their base counterparts: qwen3b shows +2.3 points and qwen7b shows +1.8 points. This gap arises because instruction-tuned models adhere more

faithfully to structured prompting and tool-use conventions, which our method relies on to compose queries, filter evidence, and update intermediate states. Base models are less consistent and therefore perform worse. For fairness, we avoid cold-start SFT and prompt-engineering, which could increase base-model performance. The gap reflects native capability.

## 4.4 QUALITATIVE ANALYSIS

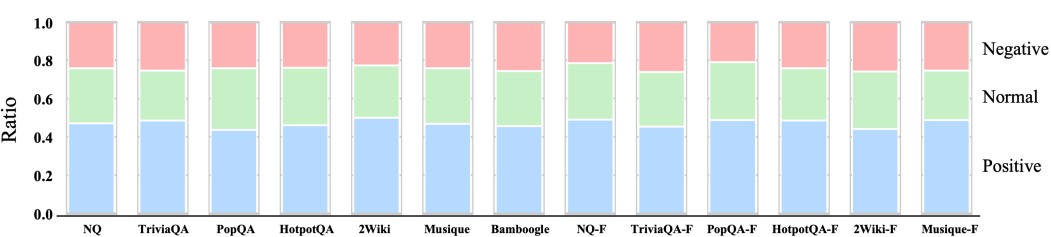

Figure 6: **Qualitative analysis of our JUDGE action impact.** We categorize each case as 'Positive' (beneficial intervention), 'Negative' (detrimental intervention), or 'Normal'.

To provide a deeper, qualitative understanding of our judge mechanism's effectiveness beyond aggregate scores, we conducted a fine-grained analysis of its behavior on a case-by-case basis. We classify the impact of each judge intervention into one of three categories, as shown in Figure 6:

**Positive (Blue):** This category represents cases where the judge provides a clear benefit. It includes two scenarios: (1) the judge correctly gives a yes signal for a state that leads to a correct final answer, and (2) the judge correctly gives a no signal for a state where the retrieved information does not contain the ground truth, effectively preventing the agent from being misled by irrelevant or distracting context. **Negative (Red):** This category captures detrimental interventions. It specifically refers to scenarios where the judge gives a yes signal, indicating the retrieved information contains the ground truth, yet the model still fails to produce the correct answer. This represents a failure case where the judge's approval did not translate to success. **Normal (Green):** All other scenarios fall into this category, representing instances where the judge's impact was neutral or ambiguous.

As illustrated in the Figure 6, the proportion of Positive outcomes is substantial across all twelve benchmark settings, consistently accounting for 40-50% of all cases. In stark contrast, the Negative outcomes constitute the smallest fraction, typically remaining below 25%. This wide gap between positive and negative impacts strongly validates our design. This qualitative evidence confirms that our judge is a reliable and highly beneficial component of the overall framework.

## 5 CONCLUSION

In this paper, we introduced ReSeek, a self-correcting framework that enables search agents to recover from intermediate reasoning errors. ReSeek equips agents with a dynamic self-correction mechanism centered on a JUDGE action, allowing them to pause, evaluate evidence, and adapt their strategy mid-episode. This process is guided by a dense, instructive reward that provides fine-grained feedback on both factual correctness and contextual utility. To enable rigorous, contamination-resistant assessment, we also introduced FictionalHot, a benchmark built around fictional entities that tests procedural reasoning over memorization. Extensive experiments show that ReSeek outperforms SOTA baselines across diverse open-domain QA tasks, with gains in multi-hop settings. Qualitative analyses confirm that the JUDGE mechanism consistently delivers substantial benefits while incurring minimal side effects, underscoring its reliability.

Beyond this, we advocate for Hot Benchmark, an evaluation principle to address inconsistencies in experimental settings that currently hinder robust comparisons. Hot Benchmark specifies a disciplined protocol for corpora, test sets, and metrics, with FictionalHot serving as a contamination-resistant stress test. We hope this principle will be adopted by the community to establish a more reproducible, transparent, and comparable foundation for measuring progress in search agents.

## 6 REPRODUCIBILITY STATEMENT

To ensure the full reproducibility of our research, we provide the complete source code for our **Re-Seek** framework and our newly constructed **FictionalHot** benchmark, both of which are accessible at `https://anonymous.4open.science/r/Re-Search-5A0F`. Our implementation is built upon standard deep learning and reinforcement learning libraries. All experiments were conducted using the base Large Language Models (LLMs) detailed in our experiment section. To facilitate verification, we specify all key hyperparameters used for training and include the exact prompts and configurations for evaluation in Appendix. We are confident that these resources provide a comprehensive basis for replicating our results and enabling further research.

## 7 ETHICAL STATEMENT

This research adheres to ethical principles for responsible AI development. We have considered the potential societal impacts of our work and have taken deliberate steps to mitigate foreseeable risks.

### 7.1 DATA USAGE AND CONTENT

Our newly introduced benchmark, **FictionalHot**, is composed of questions about fictional entities curated from publicly available web sources. This design choice inherently avoids the inclusion of Personally Identifiable Information (PII) or sensitive private data. The other established datasets used in our study are publicly available for academic research, and we have strictly adhered to their licensing terms. To the best of our knowledge, these datasets do not contain PII, and no new data collection involving human subjects was performed for this work.

### 7.2 POTENTIAL FOR MISUSE AND BIAS

We acknowledge that sophisticated search agents could potentially be misused for malicious purposes, such as generating misinformation at scale. However, the primary motivation of our research is to improve the reliability and factual grounding of these agents. The core of **ReSeek**—its self-correction mechanism and its reward function focused on correctness and utility—is designed to make agents more faithful and less prone to factual hallucination, contributing to a safer and more trustworthy AI ecosystem. Furthermore, we recognize that the LLMs used in our framework may inherit and amplify existing societal biases from their training corpora. While a comprehensive audit of all possible biases is beyond the scope of this work, we release our code with the explicit recommendation that any downstream applications undergo rigorous bias and safety testing before deployment.

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

## A  APPENDIX

### A.1  IMPLEMENTATION DETAILS

We provide a detailed description of our implementation to ensure the reproducibility of our results. Our experiments are built upon the internal `verl` reinforcement learning framework and executed on a cluster equipped with Huawei Ascend NPUs.

**Model and Data**    The core of our agent is the `Qwen2.5-3B-Instruct` model, which serves as a shared backbone for both the policy and value networks. To manage memory consumption during training, we enable gradient checkpointing. The agent was trained on the `hot_benchmark` dataset, which is formatted to match the structure of Natural Questions (NQ). For data processing, we set the maximum prompt length to 2048 tokens, the maximum response length for generation to 500 tokens, and the maximum observation length from the environment to 500 tokens.

**Training Algorithms**    In our experiments, we compared two policy optimization algorithms: the standard Proximal Policy Optimization (PPO) and Generalized Reinforcement Policy Optimization (GRPO). To ensure a fair comparison, both algorithms were trained under an identical hyperparameter configuration. The models were trained for a single epoch. The optimizer was configured with a learning rate of `1e-6` and a learning rate warmup ratio of 0.285. For policy updates, we used a training batch size of 512 episodes. This batch was processed using PPO mini-batches of size 256, which were further divided into micro-batches of size 64. To stabilize training and prevent the policy from deviating excessively from the reference model, we incorporated a KL divergence penalty with a coefficient ($\beta$) of `0.001`, calculated using the `low_var_kl` formulation. For credit assignment, we used a discount factor ($\gamma$) of 0.99 and Generalized Advantage Estimation (GAE) with a $\lambda$ of 0.95. During the rollout phase, a temperature of 1.0 was used for action sampling.

**System and Environment**    Our implementation relies on PyTorch and utilizes the `vllm` library for efficient inference during rollouts. We employed Fully Sharded Data Parallelism (FSDP) with parameter offloading to effectively distribute the model across multiple NPU devices. The experimental environment was configured with a maximum of 1 turn per episode (`max_turns=1`). The agent interacts with an external retriever service via an HTTP API, which returns the top 3 (`topk=3`) most relevant documents for a given query.

### A.2  PPO VS. GRPO AND BASE VS. INSTRUCT

**Base vs. Instruction-Tuned Backbones.**    As discussed in the main body, the choice of backbone model significantly impacts performance. The results in Table 5 confirm that instruction-tuned models generally outperform their base counterparts on most standard QA datasets. This advantage arises because our method relies on the model's ability to interpret structured prompts for composing queries, filtering evidence, and updating states. Instruction-tuned models, having been trained to follow complex instructions, adhere to these conventions more faithfully. In contrast, base models exhibit less consistency in following the structured format, leading to degraded performance.

**Superiority of GRPO over PPO.**    The empirical results also reveal a clear and consistent advantage of GRPO over PPO. This performance gap is not merely incremental but is rooted in a fundamental training stability issue we encountered with PPO. Specifically, when training with PPO, the

Table 5: Performance comparison of ReSeek trained with GRPO versus PPO on both base and instruction-tuned models. GRPO consistently outperforms PPO across most datasets and model configurations. The 'Avg.' column is the average score across all eight datasets.

| Method | NQ | TriviaQA | PopQA | HotpotQA | 2wiki | Musique | Bamboogle | FictionalHot | Avg. |
|---|---|---|---|---|---|---|---|---|---|
| **Qwen2.5-7B-Base/Instruct** | | | | | | | | | |
| ReSeek-base (GRPO) | 0.4654 | 0.60 | 0.4917 | 0.358 | 0.345 | 0.140 | 0.371 | 0.052 | 0.353 |
| ReSeek-instruct (GRPO) | **0.469** | **0.640** | **0.501** | **0.389** | **0.382** | **0.185** | **0.392** | **0.061** | **0.377** |
| ReSeek-base (PPO) | 0.391 | 0.565 | 0.418 | 0.320 | 0.317 | 0.112 | 0.345 | 0.388 | 0.357 |
| ReSeek-instruct (PPO) | 0.432 | 0.610 | 0.473 | 0.365 | 0.358 | 0.159 | 0.366 | 0.055 | 0.352 |
| **Qwen2.5-3B-Base/Instruct** | | | | | | | | | |
| ReSeek-base (GRPO) | **0.421** | **0.560** | 0.425 | 0.273 | 0.275 | 0.081 | 0.280 | **0.050** | 0.296 |
| ReSeek-instruct (GRPO) | 0.415 | 0.553 | **0.434** | **0.328** | **0.298** | **0.103** | **0.304** | 0.059 | **0.312** |
| ReSeek-base (PPO) | 0.362 | 0.495 | 0.381 | 0.275 | 0.251 | 0.065 | 0.255 | 0.291 | 0.297 |
| ReSeek-instruct (PPO) | 0.385 | 0.525 | 0.410 | 0.301 | 0.277 | 0.088 | 0.281 | 0.052 | 0.290 |

agent frequently suffered from **policy collapse** (Yuan et al., 2025), a known challenge in reinforcement learning, especially for tasks involving long generation horizons.

This issue was particularly acute in our framework due to the long and complex Chain-of-Thought (CoT) reasoning paths. We observed that after an initial learning phase, the PPO policy would abruptly degrade, characterized by a simultaneous and rapid drop in both the reward signal and the policy's entropy. This collapse rendered the model unable to perform the task, as it began generating repetitive or nonsensical outputs. In contrast, GRPO demonstrated significantly greater training stability, successfully navigating the long CoT trajectories without collapsing and achieving steady performance gains. This inherent robustness makes GRPO a far more suitable and reliable algorithm for our complex reasoning task, explaining its superior final performance.

## A.3 MORE RESULTS ON BIGGER LLMS

To evaluate the scalability and effectiveness of ReSeek on larger and more capable language models, we extend our experiments to include Qwen3-8B and Qwen3-30B-A3B-Thinking-2507. Table 6 presents a comparative analysis of ReSeek applied to a range of models, from 3B to 32B parameters.

Table 6: Performance of ReSeek on various instruction-tuned models of increasing scale.

| Method (ReSeek with) | NQ | TriviaQA | PopQA | HotpotQA | 2wiki | Musique | Bamboogle | FictionalHot | Avg. |
|---|---|---|---|---|---|---|---|---|---|
| Qwen2.5-3B-Instruct | 0.415 | 0.553 | 0.434 | 0.328 | 0.298 | 0.103 | 0.304 | 0.059 | 0.312 |
| Qwen2.5-7B-Instruct | 0.469 | 0.640 | 0.501 | 0.389 | 0.382 | 0.185 | 0.392 | 0.061 | 0.377 |
| Qwen3-8B | 0.475 | 0.635 | 0.495 | 0.401 | 0.379 | 0.192 | 0.410 | 0.058 | 0.381 |
| Qwen3-30B-A3B-Thinking-2507 | **0.495** | **0.671** | **0.521** | **0.455** | **0.458** | **0.235** | **0.560** | **0.071** | **0.433** |

The results clearly demonstrate the strong scalability of our ReSeek framework. As the model size increases, the overall performance consistently improves. We observe a significant performance leap from the 3B model (0.312 avg.) to the 7B/8B models ( 0.380 avg.), and another substantial gain with the 30B model (0.479 avg.).

Notably, the performance between Qwen2.5-7B-Instruct an Qwen3-8B is highly competitive and neck-and-neck, with each model excelling on different datasets. For instance, Qwen2.5-7B-Instruct shows a slight edge on PopQA and 2WikiMQA, while Qwen3-8B performs better on NQ and HotpotQA. This indicates that ReSeek can effectively leverage the distinct strengths of different backbone models. The Qwen3-30B-A3B-Thinking-2507 achieves the best results across almost all datasets, establishing a new level of performance.

## A.4 PERFORMANCE WITH A REAL-WORLD SEARCH ENGINE

To evaluate the real-world applicability of ReSeek, we replaced our static retrieval corpus with the Google Search API. Table 7 compares the performance of the 3B and 7B models with and without this live search capability.

Table 7: Comparison of ReSeek performance using a static retrieval corpus versus a real-world search engine (Google Search). The '(-real)' suffix denotes experiments with the Google Search API.

| Method (ReSeek with) | NQ | TriviaQA | PopQA | HotpotQA | 2wiki | Musique | Bamboogle | FictionalHot | Avg. |
|---|---|---|---|---|---|---|---|---|---|
| Qwen2.5-3B-Instruct | 0.415 | 0.553 | 0.434 | 0.328 | 0.298 | 0.103 | 0.304 | **0.059** | 0.312 |
| Qwen2.5-3B-Instruct (-real) | **0.462** | **0.605** | **0.498** | **0.371** | **0.345** | **0.138** | **0.355** | 0.055 | **0.354** |
| Qwen2.5-7B-Instruct | 0.469 | 0.640 | 0.501 | 0.389 | 0.382 | 0.185 | 0.392 | **0.061** | 0.377 |
| Qwen2.5-7B-Instruct (-real) | **0.511** | **0.695** | **0.557** | **0.442** | **0.428** | **0.224** | **0.460** | 0.059 | **0.422** |

As shown in Table 7, integrating a live search engine provides a substantial performance boost. The average score for the 7B model, for instance, increases from 0.377 to 0.422. This improvement is attributed to the access to fresher and higher-quality information from the web, which is particularly beneficial for knowledge-intensive datasets.

Crucially, performance on FictionalHot remains unaffected, as expected, since a real-world search engine contains no information about its fictional entities. This not only confirms that the performance gains stem from superior information retrieval but also demonstrates the robustness and practical utility of ReSeek in a real-world setting.

## A.5 CASE STUDY

To provide a concrete illustration of ReSeek's advantages, we present a side-by-side case study. We compare the reasoning process of ReSeek against Search-R1 on some multi-hop questions.

### A.5.1 CASE STUDY I

---

**Question:** In what year did the man who shot the Chris Stockley, of The Dingoes, die?

**Ground Truth:** 1987

**Search-R1:**

<think> To find out in what year the man who shot the Chris Stockley, of The Dingoes, died, I will search for information about the incident and the man who shot Chris Stockley. </think>

<search> In what year did the man who shot the Chris Stockley, of The Dingoes, die? </search>

<information>

Doc 1(Title: "The Dingoes") and Tolhurst although Bois' book confirms that the song was written by Tolhurst alone. A week before the single was released Stockley received a serious gunshot wound during an incident at a party in Melbourne that resulted in a two-month stay in hospital, initially described as an 'accidental shooting', according to music historian, Ian McFarlane's ""Encyclopedia of Australian Rock and Pop"", Stockley was shot by notorious drug dealer Dennis Allen, who was trying to gate crash the party. An eight-hour benefit concert was held for Stockley on 4 November 1973 at Leggett's Ballroom

Doc 2(Title: "Dennis Allen (criminal)") in 1985. One victim who survived was guitarist Chris Stockley of The Dingoes, whom Allen shot in the stomach while attempting to gatecrash a party. Allen received a ten-year prison sentence for rape during the 1970s. It is also reported that he was a major drug dealer in the Richmond and South Yarra areas of Melbourne during the 1980s.

Doc 3(Title: "The Dingoes") The Dingoes The Dingoes is an Australian country rock band initially active from 1973 to 1979, formed in Melbourne which relocated to the United States from 1976. Most stable line-up was John Bois on bass guitar, John Lee on drums, Broderick Smith on vocals and harmonica, Chris Stockley on guitar, and Kerryn Tolhurst on guitar. Mal

</information>

<answer>1985</answer>

---

Figure 7: **A baseline agent (Search-R1) failing the two-hop question.** The agent attempts to solve the problem in a single step and incorrectly extracts the year of the shooting (1985) instead of the correct year of death (1987).

---

**Question:** In what year did the man who shot the Chris Stockley, of The Dingoes, die?

**Ground Truth:** 1987

**Ours:**

<think> To find out the year the man who shot Chris Stockley, of The Dingoes, died, let's start by identifying the man who shot Chris Stockley. </think>

<search> Chris Stockley Dingoes shot</search>

<information>

Doc 1(Title: "The Dingoes") and Tolhurst although Bois' book confirms that the song was written by Tolhurst alone. A week before the single was released Stockley received a serious gunshot wound during an incident at a party in Melbourne that resulted in a two-month stay in hospital, initially described as an 'accidental shooting', according to music historian, Ian McFarlane's ""Encyclopedia of Australian Rock and Pop"", Stockley was shot by notorious drug dealer Dennis Allen, who was trying to gate crash the party.

Doc 2(Title: "The Dingoes") The Dingoes The Dingoes is an Australian country rock band initially active from 1973 to 1979, formed in Melbourne which relocated to the United States from 1976. Most stable line-up was John Bois on bass guitar, John Lee on drums, Broderick Smith on vocals and harmonica, Chris Stockley on guitar, and Kerryn Tolhurst on guitar. Mal Logan (who provided keyboards on the first LP) on keyboards joined after Stockley was hospitalised when shot in the stomach by Melbourne drug-dealer,

Doc 3(Title: "Dennis Allen (criminal)") in 1985. One victim who survived was guitarist Chris Stockley of The Dingoes, whom Allen shot in the stomach while attempting to gatecrash a party. Allen received a ten-year prison sentence for rape during the 1970s. It is also reported that he was a major drug dealer in the Richmond and South Yarra areas of Melbourne during the 1980s. New South Wales Police Detective-Sergeant Roger Rogerson was convicted of supplying heroin in a deal with Allen, but was acquitted following appeal. Allen avoided capture and prosecution for his crimes by acting as an informant for the police. Allen died on

</information>

<judge>Yes</judge>

Based on the information provided, the man who shot Chris Stockley, of The Dingoes, is Dennis Allen. Now, let's find out the year he died.

<search>Garajonay National Park location</search>

<information>

Doc 1(Title: "Dennis Allen (criminal)") Dennis Allen (criminal) Dennis Bruce Allen (7 November 1951 – 13 April 1987) was an Australian drug dealer who was reported to have murdered many victims. He was based in Melbourne, and was the oldest son of criminal matriarch Kath Pettingill. Allen avoided capture for drug dealing by acting as an informant throughout his career. He died of heart disease in 1987

Doc 2(Title: "Dennis Allen (criminal)") 13 April 1987 of heart failure at St. Vincent's Hospital, Melbourne; ""pieces of his heart actually broke off after decades of heavy drug abuse"". His funeral was conducted by Father Peter Norden, a Jesuit priest who performed funerals for three members of the Pettingill family during the 1980s. Dennis Allen (criminal) Dennis Bruce Allen (7 November 1951 – 13 April 1987)

Doc 3(Title: "Dennis Allen (criminal)") in 1985. One victim who survived was guitarist Chris Stockley of The Dingoes, whom Allen shot in the stomach while attempting to gatecrash a party. Allen received a ten-year prison sentence for rape during the 1970s. It is also reported that he was a major drug dealer in the Richmond and South Yarra areas of Melbourne during the 1980s.

</information>

<judge>Yes</judge>

Based on the information provided, Dennis Allen died on 13 April 1987. Therefore, the year the man who shot Chris Stockley, of The Dingoes, died is 1987.

<answer>1987</answer>

---

Figure 8: **A case study of ReSeek on a two-hop question.** The agent first identifies the shooter ("Dennis Allen") and then finds his death year. The judge action is used to validate the intermediate finding before proceeding to the second reasoning step.

This side-by-side comparison highlights the value of ReSeek's self-correction. The baseline agent (Figure7), treating the query as a single task, incorrectly extracts a distractor date (1985, the year of the shooting) and fails. In contrast, ReSeek's trajectory (Figure8) demonstrates robust multi-step reasoning. It first identifies the shooter ("Dennis Allen"), then uses the judge action to validate this intermediate result before initiating a second, focused search for the death year. This structured process prevents premature conclusions and leads to the correct answer where the baseline fails.

### A.5.2 CASE STUDY II

The contrast between ReSeek's success (Figure10) and the baseline's failure (Figure9) on this second case study is even more telling. The baseline agent correctly identifies the creator's name, "Loren Bouchard," but its monolithic reasoning process stops there. Unable to find the birth date in the initial context and lacking a mechanism to initiate a follow-up query, it resorts to hallucinating an incorrect answer ("1985") that is entirely absent from the evidence.

In stark contrast, ReSeek demonstrates the power of structured self-correction. After its initial search, the judge action correctly identifies that the answer has not been found (`<judge>` No `</judge>`No). This crucial validation step prompts the agent to formulate a new plan: use the newly found entity, "Loren Bouchard," to perform a second, targeted search. This methodical decomposition of the problem allows ReSeek to navigate the multi-hop query, successfully retrieve the correct birth date, and avoid the pitfall of ungrounded generation that caused the baseline to fail.

---

**Question:** When was the creator of Saddle Rash born?

**Ground Truth:** October 10, 1969

**Search-R1:**

<think> To find out when the creator of Saddle Rash was born, I will search for information about the creator. </think>

<search> When was the creator of Saddle Rash born? </search>

<information>

Doc 1(Title: "Saddle Rash") Saddle Rash Saddle Rash is a canceled comedy animated series. The pilot episode was featured on March 24, 2002 on Cartoon Network's ""Adult Swim"" programming block. It was created by Loren Bouchard, co-creator of ""Home Movies"", and uses the same low-budget Flash animation technique found in seasons two and up of ""Home Movies"". ""Saddle Rash"" is based on a small town in the Old West, and narrated by an ""old prospector""-type chara

Doc 2(Title: "Saddle sore") Saddle sore A saddle sore in humans is a skin ailment on the buttocks due to, or exacerbated by, horse riding or cycling on a bicycle saddle. It often develops in three stages: skin abrasion, folliculitis (which looks like a small, reddish acne), and finally abscess. Because it most commonly starts with skin abrasion, it is desirable to reduce the factors

Doc 3(Title: "Saddle Rash") town for the purpose of revenge against Tommy Morgan (also voiced by Benjamin), who has been hiding in the hills due to a $1000 bounty on his head. While Slim is in town, Hanna Headstrong (Sarah Silverman), the daughter of a rancher in town, falls in love with him. Eventually, Slim and Tommy Morgan stand off. ""All the members are listed in credited order."

</information>

<answer>1985</answer>

---

Figure 9: **A baseline agent (Search-R1) failing the two-hop question.** While the agent's search successfully identifies the creator, "Loren Bouchard," it fails to perform the necessary follow-up search for their birth date. It prematurely concludes with a hallucinated and incorrect answer.

**Question:** When was the creator of Saddle Rash born?

**Ground Truth:** October 10, 1969

**Search-R1:**

<think> To find out when the creator of Saddle Rash was born, let's start by identifying who the creator is. </think>

<search> creator of Saddle Rash birth date </search>

<information>

Doc 1(Title: "Saddle Rash") Saddle Rash Saddle Rash is a canceled comedy animated series. The pilot episode was featured on March 24, 2002 on Cartoon Network's ""Adult Swim"" programming block. It was created by Loren Bouchard, co-creator of ""Home Movies"", and uses the same low-budget Flash animation technique found in seasons two and up of ""Home Movies"". ""Saddle Rash"" is based on a small town in the Old West, and narrated by an ""old prospector""-type chara

Doc 2(Title: "Saddle sore") Saddle sore A saddle sore in humans is a skin ailment on the buttocks due to, or exacerbated by, horse riding or cycling on a bicycle saddle. It often develops in three stages: skin abrasion, folliculitis (which looks like a small, reddish acne), and finally abscess. Because it most commonly starts with skin abrasion, it is desirable to reduce the factors

Doc 3(Title: "Saddle Rash") town for the purpose of revenge against Tommy Morgan (also voiced by Benjamin), who has been hiding in the hills due to a $1000 bounty on his head. While Slim is in town, Hanna Headstrong (Sarah Silverman), the daughter of a rancher in town, falls in love with him. Eventually, Slim and Tommy Morgan stand off. ""All the members are listed in credited order."

</information>

<judge>No</judge>

The information provided does not directly answer the question about the birth date of the creator of Saddle Rash. Let's try another search.

<search> Loren Bouchard birth date </search>

<information>

Doc 1(Title: "Loren Bouchard") and the second began, critics began giving the series praise. The show has generally been viewed as a spiritual successor to ""King of the Hill"", which carried less emphasis on shock comedy and focused more on character-driven humor. A film adaptation based on the animated TV series is in the works and is scheduled for a July 17, 2020 release. Bouchard lives in Los Angeles with his wife, Holly Kretschmar. The couple married on September 3, 2006.

Doc 2(Title: "Sylvain Bouchard") Sylvain Bouchard Sylvain Bouchard (born April 12, 1970 in Loretteville, Quebec City, Canada) is a Canadian long track speed skater. He won the 1000m event at the 1998 World Single Distance Championships. He competed at the 1994 Winter Olympics, finishing 4th at the 500m event and 5th at the 1000m event. He also competed at the 1998 Winter Olympics, finishing 4th on 500m and 5th on 1000m. He earned the world record time in the 1000m event in 1995 and 1998.

Doc 3(Title: "Sylvain Bouchard") Sylvain Bouchard Sylvain Bouchard (born April 12, 1970 in Loretteville, Quebec City, Canada) is a Canadian long track speed skater. He won the 1000m event at the 1998 World Single Distance Championships. He competed at the 1994 Winter Olympics, finishing 4th at the 500m event and 5th at the 1000m event. He also competed at the 1998 Winter Olympics, finishing 4th on 500m and 5th on 1000m. He earned the world record time in the 1000m event in 1995 and 1998.

</information>

<judge>Yes</judge>

<answer> October 10, 1969 </answer>

Figure 10: **A case study of ReSeek on a two-hop question.** The agent first attempts a broad search but correctly uses the judge action to determine the retrieved information is insufficient. It then extracts the creator's name ("Loren Bouchard") from the initial context and initiates a second, focused search for the birth date.

