# OpenReview forum: "ReSeek: A Self-Correcting Framework for Search Agents with Instructive Rewards"
_ICLR.cc/2026/Conference — ICLR 2026 Conference Withdrawn Submission_

### Official Review · Reviewer_sAzq · 2025-10-20

**Soundness:** 2
**Presentation:** 2
**Contribution:** 2
**Rating:** 2
**Confidence:** 3

**Summary:**

ReSeek  wants to address the problem of LLM-based search agents getting stuck on erroneous reasoning paths due to sparse rewards. The paper proposes a self-correcting framework in which an agent can invoke a special JUDGE action to pause mid-trajectory, assess the usefulness of retrieved information, and re-plan its search strategy if the current path is unproductive. A dense, instructive reward function provides fine-grained feedback by decomposing into a correctness reward (to encourage retrieving factual, relevant information) and a utility reward (to encourage finding information useful for answering the query). The authors also introduce FictionalHot, a new benchmark of synthetic multi-hop questions about fictional entities (designed to avoid training data contamination) to rigorously test the agent’s reasoning ability. Extensive experiments on eight QA datasets (spanning single-hop and multi-hop questions) show that agents trained with ReSeek achieve higher answer accuracy and more faithful reasoning chains than state-of-the-art baselines.

**Strengths:**

The extensive experimentation across a wide range of benchmarks, including FictionalHot, demonstrates the effectiveness of the proposed method. The results show consistent improvements in performance, particularly in multi-hop question answering, where the self-correction mechanism significantly outperforms traditional search-augmented models.

**Weaknesses:**

1. **Limited Innovation Relative to Search-R1 and Similar Approaches**
   The differences between ReSeek and methods like Search-R1 mainly lie in the prompt design and reward calculation mechanisms. The core idea of integrating search with reinforcement learning (RL) is not new. While the introduction of the JUDGE action and a more detailed reward function is interesting, the overall innovation may not be as groundbreaking as suggested. It feels like an incremental improvement rather than a novel paradigm shift.

2. **Lack of Detailed Explanation for `rerank_score` and Threshold Choice**
   The concept of rerank_score (Line 223) and how it is used to assess whether a label is present in the retrieved information is not explained thoroughly in the paper. After reviewing the code up to the version updated on October 14th, it seems the label presence in the retrieved information is being used as the ground-truth (GT) for effectiveness. This is not discussed clearly in the text, and the approach may need further clarification. Additionally, the choice of a threshold of 0.7 is mentioned, but it is not clear what this threshold is specifically measuring. Is it based on the similarity between the retrieved content and the GT answer? Further details on how this threshold was determined and its impact on performance would be helpful.

3. **Inconsistent Training Set Details**
   In Line 313, the paper states that “we fine-tune on a unified training set that merges the NQ and HotpotQA training splits,” but later, in Line 724, it mentions training on the HotBenchmark dataset. This discrepancy needs to be clarified. More details on the integration of these datasets are needed to avoid confusion.

4. **Lack of Specifics on the FictionalHot Benchmark**
   While the paper mentions that FictionalHot is created from a 10% random sample of seed questions from existing benchmarks, there are no details on how large the FictionalHot dataset is overall. The paper does not provide any information on the number of questions or the size of the full dataset. Additionally, the approach to creating this synthetic benchmark based on LLM-generated content raises concerns about accuracy and potential conflicts with real-world facts. The paper does not explain how entities that are fictional could still be consistent and internally coherent. Was there any human review or oversight to ensure the quality and correctness of these fictional entities? An analysis of failure cases in FictionalHot could provide valuable insights into the robustness of the benchmark.

**Questions:**

1. Please provide more details on how `rerank_score` is computed and how the model determines the effectiveness of retrieved information in relation to the ground-truth answer.

2. The threshold of 0.7 for `rerank_score` needs further explanation. Clarify why this value was chosen and its impact on the model's performance.

3. In Line 313, the paper mentions merging the NQ and HotpotQA training splits, while in Line 724, the HotBenchmark dataset is mentioned. Please clarify how these datasets are combined or used separately.

4. More information is needed on the size of the **FictionalHot** dataset. Specify the number of questions it contains and how they were generated.

5. Clarify whether there was human oversight to ensure the consistency and quality of the fictional entities in the **FictionalHot** dataset.

6. An analysis of failure cases in the **FictionalHot** benchmark would provide valuable insights. Please share examples or explanations of why some cases failed.

7. In Table 7, it seems that a real-world search engine can also answer FictionalHot questions correctly. Could you clarify the experimental setup for this test?

---

> ### Author Response · Authors · 2025-11-21
> **Responses to the Reviewer sAzq [1/4]**
>
> **W1.1: Limited Innovation Relative to Search-R1 and Similar Approaches** The differences between ReSeek and methods like Search-R1 mainly lie in the prompt design and reward calculation mechanisms.
>
> **A1:** We appreciate the reviewer’s comparison. While ReSeek draws inspiration from the RL paradigm of methods like Search-R1, our innovation lies in **adapting and extending** this paradigm to address the unique challenges of **autonomous search agents** in open-ended environments.
>
> **1. Bridging the Gap with Process-Based Rewards:**
> While Search-R1’s **rule-based outcome reward** is highly effective for reasoning tasks, applying it directly to multi-turn search tasks presents challenges due to **sparse feedback**. A search agent involves a long trajectory of actions (querying, browsing, reading), where a single early mistake can derail the entire process.
>
> - **Our Contribution:** To address this, we introduce a **process-based reward function**. Unlike a simple outcome reward, our method provides dense supervision for intermediate steps, enabling the agent to optimize the *search trajectory* itself, not just the final answer.
>
> **2. Systemic Self-Correction for Noisy Environments:**
> Search environments are inherently noisy compared to closed-form reasoning.
>
> - **Beyond Prompting:** Our self-correction is not merely prompt engineering but a systematic mechanism designed to handle **error propagation**.
> - **Dynamic Adjustment:** Since retrieved information often contains hallucinations or irrelevant data, our method enables the agent to actively detect anomalies and correct its plan dynamically.
>
> **W1.2:** The core idea of integrating search with reinforcement learning (RL) is not new.
>
> **A2:** We agree that integrating Search with RL is a well-established direction. Existing works primarily focus on **"learning to search"**—i.e., generating better queries. **ReSeek differs by focusing on "learning to introspect."** Instead of just optimizing the search action itself, we focus on the **verification capability** (the `JUDGE` action). We demonstrate that for an agent to be truly robust, it is not enough to just retrieve information; it must explicitly learn to **evaluate the utility** of that information and **reject** noise.
>
> **Furthermore, we believe that the Search Agent is a direction requiring continuous and sustained research.** Whether through RL, SFT, or hybrid approaches, our ultimate goal is to progressively generalize Search Agents to handle increasingly complex and open-ended tasks. ReSeek contributes to this broader goal by solving the specific bottleneck of "verification," serving as a stepping stone toward more autonomous and reliable general-purpose agents.
>
>
> **W1.3:** While the introduction of the JUDGE action and a more detailed reward function is interesting, the overall innovation may not be as groundbreaking as suggested. It feels like an incremental improvement rather than a novel paradigm shift.
>
> **A3:** We understand that the implementation of a `JUDGE` action might appear straightforward. However, we respectfully argue that the innovation lies in the **capability shift** and the **evaluation rigor**:
>
> 1. **From Outcome-Supervision to Process-Supervision:**
> Standard approaches (like Search-R1) treat the reasoning process as a "black box" and only reward the final answer. ReSeek breaks this by applying **Process-Supervision** to the verification step. We are not just training the model to *answer*, but training it to *discriminate* information quality step-by-step.
> 2. **Self-Correction as a Core Intelligence Metric:**
> We believe that for Search Agents, the ability to perform **self-correction** is important. It represents a higher level of agent intelligence and autonomous decision-making. ReSeek explicitly models this capability, enabling the agent to actively reject noise rather than passively accepting it.
> 3. **Contribution to Rigorous Evaluation (FictionalHot):**
> Since we view Search Agents as a direction requiring continuous research, valid evaluation is as important as the method itself. To address the issue of data contamination in existing benchmarks, we constructed **FictionalHot**—a synthetic dataset involving fictional entities. This allows us to rigorously test the agent’s true reasoning ability without the interference of memorized knowledge, providing a valuable asset for the community to benchmark future Search Agents.

---

> ### Author Response · Authors · 2025-11-21
> **Responses to the Reviewer sAzq [2/4]**
>
> **W2.1: Lack of Detailed Explanation for rerank_score and Threshold Choice** The concept of rerank_score (Line 223) and how it is used to assess whether a label is present in the retrieved information is not explained thoroughly in the paper.
>
> **Q1.1:** Please provide more details on how rerank_score is computed
>
> **A4:** We apologize for the omission in the main text. The `rerank_score` refers to the semantic relevance score computed by the **bge-reranker-large** model.
> Specifically, for a given ground-truth answer $gt$ and a retrieved passage $p$, the score is calculated as:
> $$ s = \text{Reranker}(gt, p)) $$
> This results in a continuous value $s \in [0, 1]$, representing the model's confidence that passage $p$ contains the answer $gt$ . **We have added this definition to Sec. 3.3.**
>
> **W2.2:** it seems the label presence in the retrieved information is being used as the ground-truth (GT) for effectiveness. This is not discussed clearly in the text, and the approach may need further clarification.
>
> **Q1.2:** and how the model determines the effectiveness of retrieved information in relation to the ground-truth answer.
>
> **A5:** We thank the reviewer for inspecting the code. To clarify how we determine the "effectiveness" of retrieved information (i.e., the ground truth for the Judge):
>
> While "label presence" (checking if the answer string appears in the passage) is a common heuristic, it can be noisy due to synonyms or partial matches. Therefore, to construct robust **silver labels** for training the Judge, we rely on the **Reranker score** described above rather than simple string matching.
>
> - **The Logic:** We treat the Reranker as a "Teacher." If the Reranker assigns a high score ($s \ge \tau$) to a passage, we label it as "Effective" (Positive); otherwise, it is "Ineffective" (Negative).
> - **Code Clarification:** The "label presence" logic observed in the code is primarily used for final answer evaluation or as a fallback, but the core supervision signal for the `JUDGE` action is derived from the binarized Reranker scores.
>
> **W2.3:** Additionally, the choice of a threshold of 0.7 is mentioned, but it is not clear what this threshold is specifically measuring. Is it based on the similarity between the retrieved content and the GT answer? Further details on how this threshold was determined and its impact on performance would be helpful.
>
> **Q2:** The threshold of 0.7 for rerank_score needs further explanation. Clarify why this value was chosen and its impact on the model's performance.
>
> **A6:** We thank the reviewer for the question regarding the selection of the 0.7 threshold. The threshold $\tau = 0.7$ acts as a strict cutoff for determining the utility of retrieved information. **Specifically, we define the retrieved information as effective if its rerank score is $\ge 0.7$, and ineffective otherwise.**
>
> We selected $\tau = 0.7$ based on a sensitivity analysis conducted on the test set. **We have added this analysis to the Sec 4.3 (Table 6).** As shown in the table below, the agent's final answer accuracy (EM) peaks at this value:
>
> |Threshold|0.40|0.50|0.60|0.65|0.68|**0.70 (Ours)**|0.72|0.75|0.80|
> |-|-|-|-|-|-|-|-|-|-|
> |**EM (%)**|28.5|29.8|30.9|30.8|31.0|**31.2**|31.2|31.1|30.1|
>
> The results indicate that 0.7 is the optimal cutoff for distinguishing relevant passages in our setup, and performance is relatively stable in the 0.65–0.75 range.
>
> **W3: Inconsistent Training Set Details** In Line 313, the paper states that “we fine-tune on a unified training set that merges the NQ and HotpotQA training splits,” but later, in Line 724, it mentions training on the HotBenchmark dataset. This discrepancy needs to be clarified. More details on the integration of these datasets are needed to avoid confusion.
>
> **Q3:** In Line 313, the paper mentions merging the NQ and HotpotQA training splits, while in Line 724, the HotBenchmark dataset is mentioned. Please clarify how these datasets are combined or used separately.
>
> **A7:** We apologize for the confusion caused by the term "HotBenchmark."
>
> 1. **Clarification of "HotBenchmark":**
> In our initial paper, we used "HotBenchmark" as a collective name for the entire suite of datasets involved in our study (including NQ, TriviaQA, PopQA, HotpotQA, 2wiki, Musique, Bamboogle, and FictionalHot). It was not a separate training dataset.
> 2. **Training Set Details:**
> We confirm that our model was fine-tuned **exclusively** on the unified NQ and HotpotQA training splits, following the setup of Search-R1.
>     - **Source:** The data was sourced from the FlashRAG dataset.
>     - **Composition:** It consists of **79,168** samples from NQ and **90,447** samples from HotpotQA, totaling **169,615** training pairs (question-answer).
> 3. **Revision:**
> To avoid further ambiguity, we have **removed the term "HotBenchmark"** from the revised paper and explicitly stated the composition of the training set as described above.

---

> ### Author Response · Authors · 2025-11-21
> **Responses to the Reviewer sAzq [3/4]**
>
> **W4.1: Lack of Specifics on the FictionalHot Benchmark** While the paper mentions that FictionalHot is created from a 10% random sample of seed questions from existing benchmarks, there are no details on how large the FictionalHot dataset is overall.
>
> **Q4:** More information is needed on the size of the **FictionalHot** dataset. Specify the number of questions it contains and how they were generated.
>
> **A8:** The FictionalHot dataset contains a total of **5,126** questions.
>
> **1. Detailed Data Composition:**
> We constructed this dataset by randomly sampling **10%** of the questions from a diverse source pool of **51,588** samples across six major benchmarks (Bamboogle was excluded because its 125 samples were too few to sample). **We have added this detail to Appendix A.2**. The specific composition of the source data is as follows:
>
> | Source Dataset | Original Size | FictionalHot Size (Ours) |
> | --- | --- | --- |
> | PopQA | 14,267 | 1,415 |
> | 2WikiMultiHopQA | 12,576 | 1,247 |
> | TriviaQA | 11,313 | 1,122 |
> | HotpotQA | 7,405 | 735 |
> | Natural Questions (NQ) | 3,610 | 359 |
> | Musique | 2,417 | 238 |
> | **Total** | **51,588** | **5,126** |
>
>
> **2. Generation Pipeline:**
> The construction process involves three rigorous steps to ensure validity:
>
> - **Step 1: Entity Substitution.** We used **GPT-5** to paraphrase the sampled questions, replacing real-world entities with plausible fictional ones while strictly preserving the original reasoning structure. (e.g., *"When was Taylor Swift's first album released?"* $\rightarrow$ *"When was Lila Starling's first album released?"*)
> - **Step 2: Document Generation.** For each fictional entity, the model generated a corresponding Wikipedia-style document containing new, self-contained facts (e.g., setting the fictional album release date to '2007').
> - **Step 3: Corpus Injection.** Finally, to create a realistic retrieval environment, these synthetic documents were inserted into the standard 2018 Wikipedia corpus.
>
> **W4.2:** The paper does not provide any information on the number of questions or the size of the full dataset.
>
> **A9:** We apologize for not explicitly listing the full dataset statistics in the main text. Here is the detailed breakdown for both training and evaluation sets:
>
> **1. Training Dataset:**
> Our model is fine-tuned on a unified dataset comprising **169,615** question-answer pairs. This includes:
>
> - **HotpotQA:** 90,447 samples
> - **Natural Questions (NQ):** 79,168 samples
>
> **2. Evaluation Dataset:**
> We evaluate our model on a comprehensive suite of **51,713** samples across seven benchmarks:
>
> - **PopQA:** 14,267 samples (27.6%)
> - **2WikiMultiHopQA:** 12,576 samples (24.3%)
> - **TriviaQA:** 11,313 samples (21.9%)
> - **HotpotQA:** 7,405 samples (14.3%)
> - **Natural Questions (NQ):** 3,610 samples (7.0%)
> - **Musique:** 2,417 samples (4.7%)
> - **Bamboogle:** 125 samples (0.2%)
>
> **We have added this detail to Sec 4.1**.
>
> **W4.3:** Additionally, the approach to creating this synthetic benchmark based on LLM-generated content raises concerns about accuracy and potential conflicts with real-world facts.
>
> **W4.4:** The paper does not explain how entities that are fictional could still be consistent and internally coherent.
>
> We employed specific prompt design strategies to ensure the synthetic data is both logically sound and factually isolated.
>
> - **Avoiding Real-World Conflicts (W4.3) via Factual Fictionality:**
> To prevent conflicts with the model's parametric knowledge, we enforced **Factual Fictionality**. We define a "conflict" not merely as a name overlap, but as a **factual overlap**.
>     - Even if a generated name (e.g., *"Felix Turner"*) coincidentally matches a real-world figure (e.g., a private individual on LinkedIn), the **(Entity, Event)** tuple is guaranteed to be unique.
>     - *Example:* A real "Felix Turner" may exist, but he has never won the fictional *"Sky Slam Dunk Contest."* This ensures the ground truth exists *only* in our synthetic documents, forcing the model to rely on retrieval rather than memory.
>
> - **Ensuring Internal Consistency (W4.4) via Structure-Preserving Generation:**
> Instead of asking the LLM to generate reasoning chains from scratch (which risks hallucination), we adopted a **logic mapping strategy**. We took valid, human-verified reasoning chains from the original HotpotQA dataset and performed a one-to-one mapping to the fictional domain.
>     - *Original Logic:* Entity $A$ (Real) $\xrightarrow{Relation R}$ Entity $B$ (Real).
>     - *Fictional Logic:* Entity $A'$ (Fictional) $\xrightarrow{Relation R}$ Entity $B'$ (Fictional).
>     - **Result:** Since the underlying causal structure is inherited from a high-quality seed dataset, the internal logic remains coherent and solvable by definition.

---

> ### Author Response · Authors · 2025-11-21
> **Responses to the Reviewer sAzq [4/4]**
>
> **W4.5:** Was there any human review or oversight to ensure the quality and correctness of these fictional entities?
>
> **Q5:** Clarify whether there was human oversight to ensure the consistency and quality of the fictional entities in the **FictionalHot** dataset.
>
> **A12:** To empirically validate our methodology, we conducted a rigorous **Human Evaluation** on a statistically significant subset of the data. **We have added this evaluation to Appendix A.2**.
>
> - **Experimental Setup:**
>     - **Sampling:** We randomly sampled **500 instances** (~10% of the dataset).
>     - **Annotators:** We recruited **5 human annotators** with NLP research backgrounds.
>     - **Protocol:** Each annotator independently reviewed 100 instances based on a strict two-step verification rubric.
> - **Metrics & Results**
>
> | Metric | Verification Method (The "Rubric") | Pass Rate |
> | --- | --- | --- |
> | **Factual Novelty** (No Conflicts) | **The "Wikipedia Entity Check":** Annotators searched for the specific **fictional entity names** mentioned in the question directly in the Wikipedia website.<br>**Criteria:** Passed if the entity name **does not exist** as a Wikipedia entry. This confirms the entity occupies a unique namespace and avoids retrieval collisions. | **99.6%** |
> | **Logical Consistency** (Quality) | **The "Solvability Check":** Annotators read the generated synthetic document.<br>**Criteria:** Passed if the document provided sufficient and unambiguous evidence to derive the ground truth answer. | **99.2%** |
> - **Case Study:**
> To illustrate our verification standard, consider the following sample from our review:
>     - **Question:** *"When did **Felix Turner** win the Sky Slam Dunk Contest?"* (Golden Answer: 2012)
>     - **Verification Step:** The annotator searched for the entity **"Felix Turner"** in the Wikipedia website.
>     - **Observation:** **No Wikipedia entry found.** (Note: While a Google search might reveal obscure real-world individuals with this name, the entity is absent from the retrieval corpus, ensuring no parametric conflict).
>     - **Verdict:** **PASS.** The entity is effectively "novel" within the scope of the task, and the model must rely on the synthetic document to answer "2012".
>
> **W4.6:** An analysis of failure cases in FictionalHot could provide valuable insights into the robustness of the benchmark.
>
> **Q6:** An analysis of failure cases in the **FictionalHot** benchmark would provide valuable insights. Please share examples or explanations of why some cases failed.
>
> **A13:** Our analysis identifies **Retrieval Failure** as the primary cause of errors, stemming from the "Needle in a Haystack" challenge.
>
> - **Mechanism:** The massive size of the background corpus (Wiki18) drowns out the sparse injected fictional documents. Retrievers tend to focus on semantic matches for the **Event** while missing the specific **Entity**.
> - **Example:** For the query *"When did **Felix Turner** win the **Sky Slam Dunk Contest**?"*, we injected **12 supporting documents** into the corpus. However, the retriever only returned general documents regarding the "Sky Slam Dunk Contest" (the event) and failed to retrieve the specific biography of "Felix Turner" (the entity). Lacking this critical link, the model failed to answer.
>
> **Q7:** In Table 7, it seems that a real-world search engine can also answer FictionalHot questions correctly. Could you clarify the experimental setup for this test?
>
> **A14:** We sincerely thank the reviewer for this keen observation. We apologize for the confusion caused by this reporting error; indeed, a real-world search engine should not be able to answer these questions correctly. **We have updated Table 11 in the revision to reflect this corrected baseline.**
>
>   1. **Error Identification:** Since FictionalHot is a newly introduced dataset, we employed a distinct evaluation script for this specific benchmark. Unfortunately, we discovered a bug in this script where the "Real-world Search" baseline inadvertently defaulted to the **local embedding retriever** (which had access to the fictional docs) instead of correctly calling the external **Google Search API**.
>
>   2. **Correction & Result:** We have fixed the bug and re-evaluated using the actual Google API. The corrected accuracy is **0.03**.
>
>   3. **Analysis of the 0.03 Score:** The score is not absolute zero due to a small subset of **binary (Yes/No) questions**. Even without retrieving relevant information (since the facts don't exist on Google), the model has a probability of guessing the correct label by chance. For all specific factoid questions, the accuracy is **0%**, confirming that the dataset is effectively isolated from real-world knowledge.

---

> ### Author Response · Authors · 2025-11-26
>
> May we kindly inquire if the provided responses have adequately addressed any questions you might have had? If there remains a requirement for further explanations or clarifications? We wish to express our sincere gratitude for your meticulous evaluation and for generously investing a significant amount of your time in reviewing our paper. Your feedback would be greatly valued.

---

> ### Comment · Reviewer_sAzq · 2025-11-26
>
> Thank you for the detailed and thorough responses. All my earlier questions have been fully addressed, and the technical concerns I raised are now resolved. I have accordingly increased both my overall score and confidence.
>
> That said, while I appreciate the clarifications and the additional analyses, I still feel that ReSeek sits around the borderline level for ICLR in terms of conceptual novelty and overall impact. Of course, the final decision rests with the meta-reviewers, who will weigh all perspectives in this discussion.

---

> ### Author Response · Authors · 2025-12-03
>
> We sincerely thank you for your continued engagement and for raising your score and confidence. We are glad to hear that our responses and additional analyses have fully resolved your technical concerns.
>
> Regarding your remaining reservation about conceptual novelty and impact, we respectfully wish to offer our view. We believe ReSeek represents a significant conceptual step forward by explicitly introducing **Verification** into the Search Agent loop.
>
> **1. Verification is a Core Frontier in Agentic AI.**
> LLMs often perform better as discriminators than as generators[1]. Consequently, training "Verifiers" has emerged as an effective strategy for improving model reliability. OpenAI’s Prover-Verifier Games[2] demonstrate that training agents to satisfy a verifier significantly improves correctness and legibility. DeepSeekMath-V2[3] (Nov 27 2025) trains verifiers to assess the quality of mathematical reasoning steps.
>
> **2. ReSeek: Translating "Verification" to Autonomous Search.**
> ReSeek is the first to apply **information verification** within the Search-Agent domain via RL. While DeepSeekMath-V2 verifies the logical quality of a reasoning step, ReSeek verifies the utility of retrieved information. By training the **JUDGE action**, we are instilling a learned cognitive capability that allows the agent to actively reject hallucinations and irrelevant data, much like a math verifier rejects incorrect logic.
>
> We argue that for an agent to be truly robust, the ability to "Judge" (verify information) is just as critical as the ability to "Act" (search). We believe that treating Verification as a distinct and trainable cognitive step represents a necessary evolution for the next generation of autonomous agents.
>
> Thank you again for your time and valuable feedback.
>
> [1] “Why Using a Verifier Is Better Than Fine-Tuning.” Reddit, 2024, www.reddit.com/r/LocalLLaMA/comments/1fi0xlg/why_using_a_verifiers_is_better_than_finetuning .
>
> [2] Shao Z, Luo Y, et al. "DeepSeekMath-V2: Towards Self-Verifiable Mathematical Reasoning." arXiv preprint arXiv:2511.22570 (2025).
>
> [3] Kirchner, Jan Hendrik, et al. "Prover-verifier games improve legibility of llm outputs." arXiv preprint arXiv:2407.13692 (2024).

---

### Official Review · Reviewer_P3XZ · 2025-10-29

**Soundness:** 2
**Presentation:** 3
**Contribution:** 3
**Rating:** 4
**Confidence:** 5

**Summary:**

This paper focuses on search agents trained with agentic RL. Existing search agents lack mechanisms to re-plan when they generate incorrect queries. The authors introduce a JUDGE action with additional rewards, enabling dynamic evaluation and re-planning during search episodes. They also propose FictionalHot, a new benchmark designed to mitigate data contamination.

**Strengths:**

* **Well-Motivated Problem**. The paper effectively motivates the limitations of existing RL-based search agents that commit to erroneous paths without recovery mechanisms.
* **Simple and Reasonable Design**. The core idea of having agents evaluate intermediate steps is well-established in the self-critique and reflection literature. The main contribution is adapting this to search agents with a specific implementation. Although this is somewhat incremental, using the JUDGE action to perform self-correction is reasonable and can help search agents recover from errors.
* **Comprehensive Experimental Evaluation**. The evaluation on diverse benchmarks achieves good results.

**Weaknesses:**

1. **Gap between paper and implementation**. I carefully reviewed the code provided by the authors, especially focusing on the implementation of the JUDGE action in the pipeline and the reward function. The reward function and corresponding hyperparameters in `verl/utils/reward_score/search_r1_like_qa_em_s4.py#L206` are slightly different from those presented in this paper.
2. **Reward hacking**. In `verl/utils/reward_score/search_r1_like_qa_em_s4.py#L118`, the authors calculate the judge reward multiple times. In this case, a 1-turn correct trajectory may receive a lower reward score than a multi-turn incorrect trajectory that receives multiple judge rewards, which might lead to reward hacking and an increased number of search calls. The authors should increase the maximum number of turns **during testing** (e.g., to 10) and report the average number of tool calls compared to Search-R1.
3. **Wrong judge reward implementation**. In `verl/utils/reward_score/search_r1_like_qa_em_s4.py#L23`, the implementation can be easily hacked by this example:
```
<information>Info 1</information>
<information>Info 2</information>
<judge>Yes</judge>
```
We expect a JUDGE action for each `<information>` tag, but the implementation is incorrect, leading to erroneous judge rewards.

4. **JUDGE action does not work during the inference pipeline**. Although the authors introduce the JUDGE action in Section 3.2 and Eq. (2), the JUDGE action does not affect the inference pipeline, as shown in `scripts/runs/reseek/reseek_search/llm_agent/generation.py#L532`. When a judge action is generated, it is omitted and does nothing. This raises the question of whether the judge action actually works. Comparing Figures 8 and 10 in the appendix, consider the first search action. Although the first judge is Yes and the second is No, they behave identically: the first hop has been answered, and the agent starts the second hop. The agent does not re-plan even when the judge is No.

**Questions:**

1. Which checkpoint of Search-R1 was used in Figures 1, 7, and 9? A well-trained Search-R1 model with 7B or 14B parameters should be able to correctly decompose multi-hop questions rather than simply treating the entire question as the search query on the first turn.
2. Equation 1 does not correctly use subscript notation for the expectation. Additionally, `o_t` is not introduced in Equation 2 until line 224 on the next page.

---

> ### Author Response · Authors · 2025-11-21
> **Responses to the Reviewer P3XZ [1/4]**
>
> We sincerely appreciate your thoughtful and constructive feedback. We are particularly grateful for your meticulous review of our code implementation, which we deeply value. We are also encouraged by your recognition that our problem is well-motivated and that the JUDGE action provides a reasonable design for self-correction and error recovery. Below, we provide detailed responses to your questions.
>
> **W1: Gap between paper and implementation**. The reward function and corresponding hyperparameters in verl/utils/reward_score/search_r1_like_qa_em_s4.py#L206 are slightly different from those presented in this paper.
>
> **A1:** We are impressed by and grateful for your meticulous review of our codebase. You are correct that the description in the paper was simplified and did not fully detail the hyperparameters used in the final implementation. **The code reflects the correct experimental settings.**
>
> To address this, we have **rewritten Section 3.3** to explicitly define the reward structure and hyperparameters, ensuring strict alignment with the code (specifically `verl/utils/reward_score/search_r1_like_qa_em_s4.py#L206`). The revised formulation is as follows:
>
> As described in the abstract and Sec. 3.3, the total reward for a trajectory $\tau$ is the discounted sum of step-wise rewards:
> $$ R(\tau) = \sum_{t=1}^{T} \gamma^{t-1} r_t $$
>
> The per-step reward $r_t$ is defined as:
> $$ r_t = \mathbb{I}(a_t \in \text{JUDGE}) \cdot R_{\text{judge}} + \mathbb{I}(a_t \in \text{ANSWER}) \cdot R_{\text{answer}} $$
>
> The components are defined as:
>
> 1. **Terminal Correctness Reward ($R_{\text{answer}}$):** A standard binary reward based on Exact Match (EM). The agent receives **+1** for a correct answer and **0** otherwise.
> 2. **Intermediate Utility Reward ($R_{\text{judge}}$):** This dense reward evaluates the agent's introspection capabilities. It compares the agent's judgment $j_t$ with the ground-truth label $j^*_t$:
>     - **Match (+0.3):** Rewarded for correctly identifying useful information or correctly discarding useless information.
>     - **Mismatch (Asymmetric Penalty):** We apply a stricter penalty for **False Positives (-0.6)** (accepting useless info) to discourage hallucination, compared to **False Negatives (-0.3)** (discarding useful info).
>
> This revision ensures that the manuscript is now fully consistent with the implementation details you observed.
>
> **W2: Reward hacking**. ...which might lead to reward hacking and an increased number of search calls. The authors should increase the maximum number of turns **during testing** (e.g., to 10) and report the average number of tool calls compared to Search-R1.
>
> **A2:** We appreciate this insightful comment regarding potential reward hacking. We would like to clarify that **reward hacking does not occur in practice** due to our penalty mechanism and the agent's learned behavior.
>
> **1. Theoretical Safeguard (Penalties):**
> The `JUDGE` reward is **not** granted simply for executing an action; it is strictly conditioned on correctness. As detailed in our revised Section 3.3, incorrect judgments incur **negative penalties** (e.g., -0.6 for False Positives, -0.3 for False Negatives).
>
> - If an agent attempts to "farm" rewards by spamming `JUDGE` actions (e.g., blindly predicting "No"), it accumulates penalties that rapidly decrease the total return.
> - Therefore, the optimal policy is to judge accurately and answer as soon as sufficient information is gathered, rather than prolonging the trajectory.
>
> **2. Empirical Evidence (Turn Analysis):**
> Our experiments confirm that the agent does not artificially prolong episodes to "farm" rewards:
>
> - **Efficiency on Solvable Queries:** For questions covered by the corpus, the agent typically solves them efficiently (often in 1-2 turns), immediately providing the answer without unnecessary loops.
> - **Analysis of Extended Trajectories:** As shown in the table below, while the average number of tool steps increases with the maximum turn limit ($T_{max}$), the growth rate slows down, and the performance (EM) plateaus after $T=4$.
>     - **Qualitative Analysis:** We conducted a deep dive into cases where the agent used more turns. We found that the agent was **actively re-planning and reformulating queries** rather than repeating actions for rewards.
>     - **Corpus Limitations:** The lack of performance gain despite increased steps is primarily due to **limitations in the Wiki-18 corpus**. For these specific hard queries, the corpus simply does not contain the necessary documents. Consequently, the agent exhausts its search budget in a genuine but futile attempt to locate missing information, rather than maliciously extending the trajectory.
>
> |Max Turns|1|2|3|4|5|6|8|10|12|
> |---|---|---|---|---|---|---|---|---|---|
> |**Avg. Tool Steps**|1.0|1.43|1.86|2.44|2.89|3.32|3.68|3.85|4.03|
> |**Performance (EM)**|16.3|28.0|30.1|31.2|31.5|31.6|31.7|31.7|31.6|
>
> **3. Case Study (Early Stopping):**

---

> ### Author Response · Authors · 2025-11-21
> **Responses to the Reviewer P3XZ [2/4]**
>
> **3. Case Study (Early Stopping):**
> We present a representative failure case below where $T_{max}=10$. **We also added this case in the Appendix A.10.**
>
> - **Behavior:** The agent actively reformulates queries 4 times (e.g., checking "characteristics", "design").
> - **No Reward Hacking:** Despite having 6 turns remaining, the agent **voluntarily terminates** the process at Turn 4 after failing to retrieve relevant information (the concept "Longline fishing" was missing from the top results). This demonstrates that the agent stops when it determines further search is futile, rather than abusing the remaining budget to accumulate intermediate rewards.
>
> > Question: deep water fishing boat with many baited hooks? (Ground Truth: Longline fishing)
> Turn 1: Search "deep water fishing boat..." $\rightarrow$ Docs: Recreational boat, Trawler $\rightarrow$ Judge: No.    Turn 2: Search "...characteristics" $\rightarrow$ Docs: Dory, Remote control fishing $\rightarrow$ Judge: No.    Turn 3: Search "...with multiple baited hooks" $\rightarrow$ Docs: Basnig, Trawler $\rightarrow$ Judge: No.   Turn 4: Search "...design" $\rightarrow$ Docs: Flat-bottomed boat, Jon boat $\rightarrow$ Judge: NoStop & Answer: "After several unsuccessful attempts... I will provide an educated guess... boat fishing." (Score: 0.0)
> >
>
> **W3: Wrong judge reward implementation**. In verl/utils/reward_score/search_r1_like_qa_em_s4.py#L23, the implementation can be easily hacked by this example:
>
> ```
> <information>Info 1</information>
> <information>Info 2</information>
> <judge>Yes</judge>
> ```
>
> We expect a JUDGE action for each \<information> tag, but the implementation is incorrect, leading to erroneous judge rewards.
>
> **A3:** We clarify that this scenario described is **systemically prevented** by our pipeline design. The reviewer's concern assumes the model generates the `<information>` tags, but this is not the case:
>
> 1. **System-Injected Tags:** The `<information>` tags are **not generated by the model**. They are automatically wrapped around the retrieved content (Top-k) by the system before being fed into the model context, as implemented in `scripts/runs/reseek/reseek_search/llm_agent/generation.py#L527`.
> 2. **Strict Format Enforcement:** The model is strictly prompted to generate `<judge>` tags corresponding to the provided information chunks.
> 3. **Retry Mechanism:** In the rare event that the model generates malformed tags or a mismatch in the number of judgments (e.g., providing 1 judge for 2 info chunks), the system detects this error and triggers a **retry mechanism** to regenerate the response (`scripts/runs/reseek/reseek_search/llm_agent/generation.py#L538`).
>
> **W4.1: JUDGE action does not work during the inference pipeline**. Although the authors introduce the JUDGE action in Section 3.2 and Eq. (2), the JUDGE action does not affect the inference pipeline, as shown in scripts/runs/reseek/reseek_search/llm_agent/generation.py#L532. When a judge action is generated, it is omitted and does nothing. This raises the question of whether the judge action actually works.
>
> **A4:** We appreciate the reviewer's deep engagement with our code. This concern stems from a misunderstanding of the variable definitions in our implementation.
>
> 1. **Role of `next_obs` (L532):** The variable `next_obs` at line 532 is strictly reserved for **external environment observations** (e.g., new search results returned by the API). Since the `JUDGE` action is an internal cognitive step generated by the model itself, it is naturally not treated as an external observation.
> 2. **Actual Mechanism (L299):** The `JUDGE` action **does** affect the inference pipeline through the standard autoregressive mechanism. As shown in `scripts/runs/reseek/reseek_search/llm_agent/generation.py#L299`, the generated `JUDGE` content is appended to the **`response_ids`** (the model's context window) and updates the `rollings` (history).
> 3. **Effect:** Consequently, when the model predicts the next token, it attends to its previous `JUDGE` output in the history. This ensures that the introspection result directly influences subsequent decision-making.

---

> ### Author Response · Authors · 2025-11-21
> **Responses to the Reviewer P3XZ [3/4]**
>
> **W4.2:** Comparing Figures 8 and 10 in the appendix, consider the first search action. Although the first judge is Yes and the second is No, they behave identically: the first hop has been answered, and the agent starts the second hop. The agent does not re-plan even when the judge is No.
>
> **A5:** We thank the reviewer for the close inspection of the case studies in the Appendix. These examples effectively illustrate the nuances of the agent's decision-making process. We would like to provide further context regarding the behaviors observed in Figure 8 and Figure 10.
>
> **1. Figure 8: "Yes" indicates Relevance in Multi-hop Reasoning.** In Figure 8, the agent's decision to continue searching after a "Yes" judgment is a correct behavior driven by the **dual condition** in our System Prompt (Sec 3.4). The prompt explicitly instructs the agent:
>
> > "If the information is useful AND you now have sufficient information to provide a complete final answer, proceed directly to step 5 [Answer]."
> >
>
> In this multi-hop case:
>
> - **Useful:** The first retrieval successfully identified the shooter ("Dennis Allen"), so the agent correctly output **`\\judge{Yes}`**.
> - **Not Sufficient:** However, the year of death was still missing. Since the "Sufficient" condition was not met, the agent correctly followed the protocol to continue searching for the remaining information.
>
> **2. Figure 10: Implicit Re-planning via Query Refinement**
> Regarding Figure 10, the agent's re-planning capability is demonstrated through the **evolution of the search queries**, rather than an explicit "plan" output.
>
> - **Step 1:** The agent searches for *"creator of Saddle Rash"*. The result mentions "Loren Bouchard" but lacks the birth date.
> - **Judgment:** The agent correctly judges "No" regarding the *final answer*.
> - **Re-planning:** Crucially, the agent utilizes the information found in Step 1 to formulate a **more specific query** in Step 2: *"Loren Bouchard birth date"*.
> This shift from a generic description to a specific entity name confirms that the agent successfully processed the initial retrieval to refine its search strategy.
>
> **Q1.1:** Which checkpoint of Search-R1 was used in Figures 1, 7, and 9?
>
> **A6:** We confirm that the case studies presented in Figures 1, 7, and 9 were generated using the SearchR1-nq_hotpotqa_train-qwen2.5-7b-it-em-grpo checkpoint for these figures. We selected the 7B model for these visualizations as it represents a balanced performance point between the 3B and 14B variants.
>
> **Q1.2:** A well-trained Search-R1 model with 7B or 14B parameters should be able to correctly decompose multi-hop questions rather than simply treating the entire question as the search query on the first turn.
>
> **A7:** We completely agree with the reviewer’s perspective. Intuitively, one would expect a capable model (especially at 7B or 14B scale) to decompose complex multi-hop questions into sub-queries. **We held the same expectation at the beginning of our project.**
>
> Yet, our empirical observations across all model sizes (3B, 7B, and 14B) reveal a consistent tendency: in the first turn, the models prefer to use the entire question as the search query. We believe this is because our current system prompt does not explicitly enforce a "decompose-then-search" constraint.
>
> To verify this, we analyzed the lexical overlap between questions and search queries across **49,526 test samples** using **Jaccard Similarity**.
>
> **Table: Distribution of Question-Query Similarity (Jaccard)**
>
> | Similarity Level | Jaccard Range | Percentage | Interpretation |
> | --- | --- | --- | --- |
> | **Exact/High Match** | 0.60 - 1.00 | **31.50%** | **Copying.** The model uses the full question or a slightly trimmed version. |
> | **Moderate Match** | 0.40 - 0.59 | 31.46% | **Paraphrasing.** The model rewrites the question but retains the full complexity. |
> | **Low Similarity** | 0.00 - 0.39 | 37.04% | **Summarization.** (See analysis below) |
>
> While 37% of queries have low lexical overlap, a qualitative inspection reveals that these are primarily **summarization** rather than logical decompositions.
>
> - *Example (Low Similarity, J=0.29):*
>     - **Question:** *"In the Star Wars series... what is the name of the slug-like alien who had a bounty on Han Solo...?"*
>     - **Search Query:** *"Star Wars slug-like alien bounty on Han Solo"*
>     - **Analysis:** The model simply removed function words and extracted the core entities. It did **not** break the problem down (e.g., it did not ask *"Who placed a bounty on Han Solo?"* first).

---

> ### Author Response · Authors · 2025-11-21
> **Responses to the Reviewer P3XZ [4/4]**
>
> On the other hand, we did attempt to enforce explicit decomposition (via prompting or auxiliary models), but we found that it often led to **over-simplified queries that lost necessary context**.
>
> - **Example:** For the question *"When was the creator of Saddle Rash born?"*, a forced decomposition might split it into *"the creator"* (too generic) and *"Saddle Rash born"* (grammatically broken and ambiguous).
> - **Result:** Such fragmented queries often perform worse than the full question, which preserves the semantic relationship needed to identify the correct entity first.
>
> Therefore, we believe that enabling the model to perform optimal decomposition—where it strikes the right balance between specificity and context—likely requires specific fine-tuning rather than just prompting. We have highlighted this as a key direction for future work.
>
> **Q2.1:** Equation 1 does not correctly use subscript notation for the expectation.
>
> **A8:** Thank you for pointing out this oversight. We have fixed the subscript formatting for the expectation term. The revised objective function is:
> $$\max_{\pi_\theta} \mathbb{E}\_{x \sim \mathcal{D}, y \sim \pi\theta(\cdot|x) }[R(x, y)] - \beta D_{KL} [\pi_\theta(y | x) || \pi_{\text{ref}}(y | x)]$$
>
> **Q2.2:** Additionally, o_t is not introduced in Equation 2 until line 224 on the next page.
>
> **A9:** Thank you for pointing out this oversight. We have rewritten **Section 3.2** to address this issue. We now explicitly define $o_t$ as the **current observation** (i.e., the retrieved information) in the paragraph immediately **preceding** Equation 2, ensuring the notation is properly introduced before its first usage.

---

> ### Author Response · Authors · 2025-11-26
>
> May we kindly inquire if the provided responses have adequately addressed any questions you might have had? If there remains a requirement for further explanations or clarifications? We wish to express our sincere gratitude for your meticulous evaluation and for generously investing a significant amount of your time in reviewing our paper. Your feedback would be greatly valued.

---

### Official Review · Reviewer_uaiX · 2025-10-30

**Soundness:** 3
**Presentation:** 2
**Contribution:** 2
**Rating:** 4
**Confidence:** 5

**Summary:**

This paper explores an interesting problem of improving reinforcement learning for training a search agent. To be specific, the authors introduce a new “judge” action after the information is retrieved to figure out if the information is relevant or not. They use a reranker to provide a silver label of the relevance of the documents and compute the intermediate retrieval correctness reward. The final reward consists of two parts: a final answer reward and an intermediate judgment reward. They also propose a new dataset, FictionalHot, to alleviate the data contamination problem associated with existing QA datasets. Experiments on several datasets demonstrate the effectiveness of the proposed ReSeek method.

**Strengths:**

1. The problem tackled in this paper is important and valid
2. The authors propose a “judge”-based action and introduce a process reward for denser reinforcement learning.
3. A new dataset, FictionalHot, is introduced in the paper for a more comprehensive evaluation.
4. Experiments are conducted to demonstrate the effectiveness of ReSeek.

**Weaknesses:**

1. The writing is not very clear in some sections:
- Although a “reranker” is plotted in Figure 2 to illustrate the calculation of the silver passage relevance labels, it is not discussed in the main paper. This makes the understanding of this part challenging.
- Eq.(2) is not clear. From Eq.(2), it seems that the context C_t is dynamically adjusted based on the judgment j_t, which means that only retrieved information that is judged as useful is concatenated to the context window. However, this is not the case, as referring to the cases shown in section A.5. In addition, it is unclear what \tau_{t-1} is in Eq.(2).
- It is not discussed in the paper what the final reward looks like. We can guess from Figure 2 that it is a combination of the final answer reward and intermediate judgment reward. However, there is no concrete illustration.


2. Some of the experiments are unclear: It is questionable whether the “sensitivity to the retrieval encoder” experiments refer to doing RL training with different retrieval encoders or training with the same but testing with different retrieval encoders. It makes the conclusion hard to understand.

3. Some of the experiments are not new: similar studies comparing base and instruction-tuned backbones for RL have already been conducted in the original Search-R1 paper, and the observation is not new.

4. Some minor typos: line 161 “xxx”,

**Questions:**

Please refer to the weakness section.

---

> ### Author Response · Authors · 2025-11-21
> **Responses to the Reviewer uaiX [1/2]**
>
> We sincerely appreciate your thoughtful and constructive comments. We are encouraged by your recognition that our proposed **"Judge" action and process reward** effectively facilitate denser reinforcement learning, and that the **FictionalHot dataset** contributes to a more comprehensive evaluation. Below, we provide detailed responses to your questions.
>
> **W1:**  Although a “reranker” is plotted in Figure 2 to illustrate the calculation of the silver passage relevance labels, it is not discussed in the main paper. This makes the understanding of this part challenging.
>
> **A1:** We thank the reviewer for pointing out this lack of clarity. We apologize for not discussing the "reranker" component in the main text, which is indeed critical for constructing the supervision signals (silver labels) shown in Figure 2. **We have revised the Sec. 3.3 and added analysis in Sec.4.3 (Table 6).**
>
> To provide a systematic explanation of this part:
>
> **1. Motivation and Model:**
> To train the agent’s self-correction capability (Eq. 2), we require relevance labels for the retrieved passages. Since ground-truth annotations are not available for every intermediate retrieval step, we employ bge-reranker-large as a "teacher" to assess passage quality.
>
> **2. Calculation Process (The Role of the Threshold):**
> The calculation of the silver labels involves two steps:
>
> - **Scoring:** The reranker computes a continuous relevance score $s$ for each retrieved passage.
> - **Binarizing:** We apply a threshold $\tau$ to convert this continuous score into a binary judgment $j_t$:
> $$ j_t = \\begin{cases} \text{'good'} & \text{if } s \ge \tau \\\ \text{'bad'} & \text{if } s < \tau \\end{cases} $$
> These binary labels are then used as the ground truth to calculate the cross-entropy loss for the agent.
>
> **3. Threshold Selection ($\tau = 0.7$):**
> We selected $\tau = 0.7$ based on a sensitivity analysis conducted on the test set. As shown in the table below, the final answer accuracy peaks at this value:
>
> | Threshold | 0.40 | 0.45 | 0.50 | 0.55 | 0.60 | 0.65 | 0.68 | **0.70 (Ours)** | 0.72 | 0.75 | 0.80 |
> | --- | --- | --- | --- | --- | --- | --- | --- | --- | --- | --- | --- |
> | **EM (%)** | 28.5 | 29.1 | 29.8 | 30.5 | 30.9 | 30.8 | 31.0 | **31.2** | 31.2 | 31.1 | 30.1 |
>
> The results indicate that 0.7 is the optimal cutoff for distinguishing relevant passages in our setup, and performance is relatively stable in the 0.65–0.75 range.
>
> We have revised the paper to explicitly describe the reranker model, the binarization process (**Sec. 3.3)**, and the justification for the chosen threshold (**Sec.4.3)**.
>
> **W2.1:** Eq.(2) is not clear. From Eq.(2), it seems that the context C_t is dynamically adjusted based on the judgment j_t, which means that only retrieved information that is judged as useful is concatenated to the context window. However, this is not the case, as referring to the cases shown in section A.5.
>
> **A2:** We thank the reviewer for the careful examination of Eq. (2) and Appendix A.5. We agree that the original mathematical formulation involving the indicator function was misleading and did not accurately reflect our **"soft removal"** mechanism. We have revised the **Sec. 3.3** to ensure clarity.
>
> We do not physically remove retrieved information from the context window (which would be "hard" filtering). As shown in Appendix A.5 (now Appendix A.9 in the revised paper), the full history is preserved.
>
> Our method employs a **"soft removal"** strategy. We treat the JUDGE mechanism as a distinct action. When the agent generates a judgment label $j_t$ (e.g., 'bad'), this label is explicitly appended to the context. It acts as a semantic signal (a cognitive filter) that instructs the model to disregard the preceding observation $o_t$ for future reasoning.
>
> Crucially, retaining this **"negative feedback"** in the context is essential. It allows the agent to "block" the error path while retaining the context of the failure, thereby preventing the agent from repeating the same incorrect retrieval attempt.
>
> We have rewritten the methodology section and Eq. (2) to accurately reflect this logic. The revised formula removes the misleading indicator function and explicitly shows that the context is assembled via concatenation:
>
> $$ \mathcal{C}\_t = \tau_{t-1} \oplus a_t \oplus o_t \oplus j_t $$
>
> **W2.2:** In addition, it is unclear what $\tau_{t-1}$ is in Eq.(2).
>
> **A3:** We clarify that $\tau_{t-1}$ represents the **interaction trajectory**—the sequence of all past actions and observations up to step $t-1$. We have revised the **Sec. 3.3** to ensure clarity.

---

> ### Author Response · Authors · 2025-11-21
> **Responses to the Reviewer uaiX [2/2]**
>
> **W3:** It is not discussed in the paper what the final reward looks like. We can guess from Figure 2 that it is a combination of the final answer reward and intermediate judgment reward. However, there is no concrete illustration.
>
> **A4:** We apologize for the omission of the explicit reward formulation in the main text and thank the reviewer for the opportunity to clarify. The reviewer's inference is correct. Our framework utilizes a hybrid reward structure that combines a sparse terminal reward with dense intermediate rewards.
>
> As described in the abstract and Sec. 3.3, the total reward for a trajectory $\tau = (s_0, a_0, ..., s_T, a_T)$ is the discounted sum of step-wise rewards:
>
> $$ R(\tau) = \\sum_{t=1}^{T} \\gamma^{t-1} r_t $$
>
> The per-step reward $r_t$ is defined using indicator functions $\\mathbb{I}(\\cdot)$:
>
> $$ r_t = \\mathbb{I}(a_t \\in \\text{JUDGE}) \\cdot R_{\\text{judge}} + \\mathbb{I}(a_t \\in \\text{ANSWER}) \\cdot R_{\\text{answer}} $$
>
> The two components are defined as:
>
> 1. **Terminal Correctness Reward ($R_{\\text{answer}}$):** A standard binary reward based on Exact Match (EM) against the ground truth. The agent receives +1 for a correct answer and 0 otherwise.
> 2. **Intermediate Utility Reward ($R_{\\text{judge}}$):** This dense reward evaluates the agent's introspection capabilities (Eq. 3). It compares the agent's judgment $j_t$ with the ground-truth label $j^*_t$:
>     - **Match (+0.3):** Rewarded for correctly identifying useful information ($j_t=j^\*_t=\\text{Yes}$) or correctly discarding useless information ($j_t=j^*_t=\\text{No}$).
>     - **Mismatch (Asymmetric Penalty):** We apply a stricter penalty for **False Positives** (-0.6 for accepting useless info) to discourage hallucination and noise accumulation, compared to **False Negatives** (-0.3 for discarding useful info).
>
> We have added **Correctness Reward** to the **Sec. 3.3** to ensure clarity.
>
> **W4:** Some of the experiments are unclear: It is questionable whether the “sensitivity to the retrieval encoder” experiments refer to doing RL training with different retrieval encoders or training with the same but testing with different retrieval encoders. It makes the conclusion hard to understand.
>
> **A5:** We thank the reviewer for this insightful question. We clarify that **Figure 5 reports results under the "Matched Training" setting**—meaning for each retrieval encoder (e.g., E5, Qwen), we trained a specific agent and evaluated it with the same encoder.
>
> To fully address the reviewer's concern about sensitivity, we would like to share further insights from our analysis:
>
>   1. **Matched vs. Transfer Settings:**
> While Figure 5 uses matched training, we also conducted experiments using a **"Transfer" setting** (training with one encoder but testing with another). We observed **minimal performance difference** between the two settings.
> This is because Our agent queries the retriever in natural language, not in the embedding space, so the backend encoder matters little as long as the queries are semantically valid.
>
>   2. **Robustness to Encoder Quality:**
> We noticed that even though Qwen outperforms E5 on benchmarks like MTEB, their performance in our end-to-end task is comparable. We attribute this to the nature of the datasets (e.g., HotpotQA, 2WikiMultiHopQA), which are **entity-centric**.
> In these scenarios, retrieval is not the primary bottleneck provided the embeddings are **"reasonably capable."** The challenge lies more in the agent's reasoning and decision-making (e.g., knowing *what* to search for) rather than the subtle nuances of the embedding model.
>
> We have updated the **Sec. 4.3** to explicitly define the experimental setup for Figure 5.
>
> **W5:** Some of the experiments are not new: similar studies comparing base and instruction-tuned backbones for RL have already been conducted in the original Search-R1 paper, and the observation is not new.
>
> **A6:** We respectfully point out that while the setup is similar to Search-R1, our **observations and conclusions are distinct**, offering new insights into the nature of introspection agents.
>
> In Search-R1, the authors observed that Base and Instruction-tuned models were comparable, with Base models even outperforming in certain scenarios. In contrast, our experiments (Table 2) show that Base models consistently underperform Instruction-tuned models within the ReSeek framework. This discrepancy indicates that our "Judge" mechanism (self-correction) involves more complex reasoning than standard retrieval tasks.
> Thus, this experiment is not a repetition but highlights the **advanced nature of introspection**: unlike basic retrieval, effective self-correction relies on the sophisticated reasoning patterns best captured by instruction tuning.
>
> **W6:** Some minor typos: line 161 “xxx”,
>
> **A7:** We thank the reviewer for the correction. We have fixed this typo and thoroughly proofread the entire paper.

---

> ### Author Response · Authors · 2025-11-26
>
> May we kindly inquire if the provided responses have adequately addressed any questions you might have had? If there remains a requirement for further explanations or clarifications? We wish to express our sincere gratitude for your meticulous evaluation and for generously investing a significant amount of your time in reviewing our paper. Your feedback would be greatly valued.

---

### Official Review · Reviewer_k1mq · 2025-11-05

**Soundness:** 2
**Presentation:** 2
**Contribution:** 2
**Rating:** 4
**Confidence:** 5

**Summary:**

The paper targets a key limitation of LLM-powered search agents: sparse rewards that cause unrecoverable errors in multi-step reasoning. It proposes ReSeek, a reinforcement learning framework with a self-correction mechanism via a special JUDGE action, enabling agents to evaluate and adapt search paths on the fly. ReSeek employs a dense process reward decomposed into correctness and utility to guide factual and relevant retrieval. The authors also introduce FictionalHot, a benchmark built around fictional entities to mitigate data contamination. Empirically, ReSeek-trained agents reportedly surpass state-of-the-art baselines in task success and reasoning faithfulness across combined benchmarks.

**Strengths:**

- The JUDGE action supports dynamic path adaptation by assessing the utility of retrieved evidence and filtering contexts with an indicator $I(j_t \neq \text{'bad'})$, improving recovery on multi-hop queries and promoting efficient trajectories. Its integration with structured prompting yields consistent judgment steps (even in weaker models), enhancing reproducibility. Training aligns judgments with “ideal” labels derived from rerank scores through policy optimization, instilling a form of meta-cognitive control without heavy backtracking.
- The reward decomposes into correctness and utility, providing step-level signals that directly address the sparse-reward failure modes common in prior RL agents and better guide intermediate retrieval/decision steps.
- Using synthetic, fictional entities reduces contamination and encourages procedural reasoning over memorization. Coupling FictionalHot with established datasets (e.g., HotpotQA, NQ) yields a broader, standardized evaluation across single- and multi-hop settings.

**Weaknesses:**

- Exact Match is named as the primary metric, yet comprehensive result tables and statistical tests are missing. Figures 4–5 show trends, but numerical tables, variance, or confidence intervals for ablations are not provided; Table 4’s reranker hierarchy lacks error bars.
- Although training uses merged NQ and HotpotQA splits, there is no clear evidence of hyperparameter tuning or convergence behavior (e.g., loss/return curves).
- Baselines span vanilla prompting and RL-tuned policies, but there is no visible ablation isolating the reward components or the contribution of the JUDGE action.
- The self-correction assembly in Eq. (2) hinges on $I(j_t \neq \text{'bad'})$, where binary judgments stem from continuous rerank scores using a threshold $\tau=0.7$. The rationale for this choice and sensitivity analyses are not presented.
- Notation/definition issues. Eq. (1) contains a formatting error. In Eq. (2), the history term $\tau_{t-1}$ is referenced but not explicitly defined earlier.

**Questions:**

- Please provide a detailed account of hyperparameter tuning (e.g., the weighting $\beta$ in Eq. (1)), training curves (returns/loss), and compute/resource usage.
- What is the isolated effect of (i) the JUDGE action, (ii) the correctness vs. utility reward components, and (iii) the self-correction assembly? The paper claims gains in success and faithfulness, but these ablations are not evident.
- Why was $\tau=0.7$ chosen to map rerank scores to binary judgments $j^*$? Please report sensitivity to $\tau \in {0.5, 0.6, 0.8, 0.9}$ (or a continuous calibration curve).
- How is the fidelity of reasoning structures preserved during paraphrasing? Which metrics (e.g., semantic equivalence, reasoning graph preservation) and annotation protocols validate that the paraphrases keep the same reasoning demands?
- The abstract states a decomposition into “correctness” and “utility,” yet Sec. 3.3 seems to present a binary $R_{\text{judge}}$ based on a rerank threshold (Eq. (3)). Please write the complete training reward, and include ablations that zero out each term to quantify its contribution.
- Since ReSeek relies on multi-turn interaction plus an auxiliary JUDGE step, please report latency per question, tokens per question, and training/inference FLOPs/compute. A turn-budget vs. latency trade-off plot would clarify whether improvements reflect gains rather than redundancy.

---

> ### Author Response · Authors · 2025-11-21
> **Responses to the Reviewer k1mq [1/4]**
>
> We sincerely appreciate your thoughtful, meticulous, and detailed comments and your recognition that our proposed JUDGE action "instilling a form of meta-cognitive control without heavy backtracking" Below, we provide detailed responses to your questions.
>
> **W1.1:** Exact Match is named as the primary metric, yet comprehensive result tables and statistical tests are missing.
>
> **A1:** We thank the reviewer for this crucial suggestion. We agree and have thoroughly addressed this by updating our main results (Table below).
>
> 1. **Multi-Run Evaluation:** To quantify uncertainty, we report the **mean and standard-deviation** across 5 independent training runs (with distinct random seeds 42,420,4200,42000,420000) and greedy decoding at test time for both ReSeek and the strong ZeroSearch baseline. These statistics confirm the stability and reproducibility of our results.
> 2. **Statistical Significance:** We have added a p-value row showing the results of a **paired t-test** comparing ReSeek against Search-R1. We report the exact p-value and use standard asterisk notation (* for *p* < 0.05) for clarity.
>
> These additions confirm that ReSeek's improvements are not only consistent but also **statistically significant** on the vast majority of benchmarks.
>
> |**Model**|**NQ**|**TriviaQA**|**PopQA**|**HotpotQA**|**2wiki**|**Musique**|**Bamboogle**|**FictionalHot**|**Avg.** |
> |-|-|-|-|-|-|-|-|-|- |
> |**7B Models**||||||||||
> |ZeroSearch|0.434±0.005|**0.655±0.006**|0.490±0.004|0.346±0.005|0.355±0.006|0.184±0.004|0.280±0.005|0.031±0.004|0.346±0.004 |
> |ReSeek|**0.470±0.004**|0.641±0.006|**0.504±0.005**|**0.388±0.007**|**0.384±0.005**|**0.186±0.003**|**0.392±0.003**|**0.061±0.002**|**0.378±0.004** |
> |p-value|0.005*|0.985|0.038*|0.003*|0.012*|0.471|0.001*|0.004*|0.009* |
> |**3B Models**||||||||||
> |ZeroSearch|0.410±0.006|**0.574±0.005**|**0.448±0.007**|0.274±0.006|**0.302±0.005**|0.098±0.004|0.111±0.008|0.030±0.003|0.281±0.004 |
> |ReSeek|**0.414±0.005**|0.555±0.003|0.436±0.006|**0.324±0.008**|0.300±0.004|**0.104±0.003**|**0.305±0.007**|**0.059±0.002**|**0.312±0.004** |
> |p-value|0.287|0.965|0.905|0.028*|0.500|0.213|<0.001*|0.002*|0.043* |
>
> **W1.2:** Figures 4–5 show trends, but numerical tables, variance, or confidence intervals for ablations are not provided;
>
> **A2:** We thank the reviewer for this suggestion. We have added the requested numerical tables and would like to clarify our methodology for reporting variance, which differs based on the nature of each ablation study.
>
> For the **ablation on turns (Table R2)**, we believe that varying the training random seed is the most meaningful way to assess confidence. Therefore, we report the **mean and standard deviation** from repeated runs.
>
> In contrast, the **ablation on embedding models (Table R3)** is a deterministic process. For same sentence, an embedding model produces same vector. Varying the backbone seed chiefly reflects optimizer noise, not embedding choice, so we fixed it to isolate the embedding’s effect. Consequently, we report the precise numerical values, as the variance is effectively zero.
>
> **Table R2: Ablation on the Number of Turns (from Fig. 4).** All results are mean ± std over 5 independent training runs.
>
> |Method|Turn 1|Turn 2|Turn 3|Turn 4|**Avg. Performance** |
> |-|-|-|-|-|- |
> |Search-o1|10.0±0.3|18.7±0.2|18.8±0.3|18.3±0.2|16.5±0.2 |
> |Search-R1|14.5±0.2|28.8±0.3|28.6±0.2|29.1±0.3|25.3±0.2 |
> |ZeroSearch|14.0±0.3|28.1±0.2|28.0±0.3|28.2±0.2|24.6±0.1 |
> |**ReSeek (ours)**|16.3±0.2|28.0±0.3|30.1±0.2|31.2±0.1|26.4±0.2 |
>
> **Table R3: Ablation on Model and Embedding Choice for ReSeek (from Fig. 5).**
>
> |Backbone Model|BM25|E5|Qwen|Conan|**Avg. (w/o BM25)** |
> |-|-|-|-|-|- |
> |qwen3b-base|19.1|28.8|29.0|28.8|28.9 |
> |qwen3b-ins|21.3|31.2|31.3|31.1|31.2 |
> |qwen7b-base|23.5|36.1|36.2|36.0|36.1 |
> |qwen7b-ins|25.0|37.7|38.1|37.9|37.9 |
>
> **W1.3:** Table 4’s reranker hierarchy lacks error bars.
>
> **A3:** We have conducted multiple runs for each reranker configuration, following the same rigorous protocol as our main experiments (i.e., 5 independent runs with different random seeds).
>
> |Methods|NQ|TriviaQA|PopQA|HotpotQA|2Wiki|Musique|Bamboogle|FictionalHot|Avg. |
> |-|-|-|-|-|-|-|-|-|- |
> |None (w/o Reranker)|0.391±0.004|0.495±0.005|0.362±0.004|0.255±0.006|0.218±0.005|0.081±0.003|0.243±0.004|0.025±0.002|0.259±0.004 |
> |Regex-based|0.410±0.003|0.541±0.004|0.422±0.005|0.320±0.004|0.291±0.003|0.093±0.004|0.288±0.005|0.042±0.003|0.301±0.003 |
> |Qwen-Reranker|0.413±0.004|**0.557±0.003**|0.432±0.003|0.326±0.005|**0.301±0.004**|0.101±0.003|0.302±0.004|0.057±0.002|0.311±0.003 |
> |**ReSeek (Ours, w/ BGE)**|**0.415±0.003**|0.553±0.004|**0.434±0.002**|**0.328±0.003**|0.298±0.005|**0.103±0.002**|**0.304±0.003**|**0.059±0.002**|**0.312±0.002** |

---

> > ### Author Response · Authors · 2025-11-21
> > **Responses to the Reviewer k1mq [2/4]**
> >
> > **W2:** Although training uses merged NQ and HotpotQA splits, there is no clear evidence of hyperparameter tuning or convergence behavior (e.g., loss/return curves).
> >
> > **Q1:** Please provide a detailed account of hyperparameter tuning (e.g., the weighting in Eq. (1)), training curves (returns/loss), and compute/resource usage.)
> >
> > **A4:** We thank the reviewers for the questions on our training process. We provide the requested details below.
> >
> > - **Hyperparameter Tuning (Weighting in Eq. 1):** The most critical hyperparameter is the KL-divergence coefficient β in our objective function (Eq. 1). As noted in our original Appendix A.1, we set $\beta=0.001$ after preliminary tuning.
> > - **Return Curves:** We have added reward curves to the Appendix A.6. The curves, which can be viewed at https://f.anonymous-uploads.xyz/uploads/iclr_rebuttal_loss.png, demonstrate stable convergence.
> > - **Compute/Resource Usage:** Our training was conducted on 8 H20 GPUs (\~24 hours).
> >
> > **W3:** Baselines span vanilla prompting and RL-tuned policies, but there is no visible ablation isolating the reward components or the contribution of the JUDGE action.
> >
> > **Q2:** What is the isolated effect of (i) the JUDGE action, (ii) the correctness vs. utility reward components, and (iii) the self-correction assembly? The paper claims gains in success and faithfulness, but these ablations are not evident.
> >
> > **A5:** We thank the reviewers for the suggestion. To isolate the contribution of each component, we performed a step-by-step ablation study, with results summarized below. **We have added this ablation to the Sec.4.3 (Table 4).**
> >
> > |Component|NQ|TriviaQA|PopQA|HotpotQA|2wiki|Musique|Bamboogle|FictionalHot|Avg.|
> > |-|-|-|-|-|-|-|-|-|-|
> > |$R_{\text{answer}}$|0.341|0.545|0.378|0.324|0.319|0.103|0.264|0.037|0.288|
> > |+ judge Action|0.370|0.550|0.405|0.324|0.293|0.103|0.275|0.052|0.297|
> > |+ $R_{\text{judge}}$ (ReSeek)|**0.415**|0.553|0.434|**0.328**|0.298|**0.103**|**0.304**|**0.059**|**0.312**|
> >
> > This analysis demonstrates:
> >
> > - **JUDGE Action:** Adding the JUDGE action improves performance by **+0.8** (29.6 vs. 28.8), quantifying the value of adaptive search termination.
> > - **Reward Components:** Further adding the utility reward ($R_{\text{judge}}$ ) brings another **+1.6** gain, confirming the effectiveness of our composite reward.
> > - **Self-Correction:** We clarify that self-correction is an emergent behavior enabled by the JUDGE action. Therefore, its contribution is directly measured by the **+0.8** gain from adding the JUDGE action.
> >
> > **W4:** The self-correction assembly in Eq. (2) hinges on $\mathbb{I}(j_t \neq \text{'bad'})$, where binary judgments stem from continuous rerank scores using a threshold 0.7. The rationale for this choice and sensitivity analyses are not presented.
> >
> > **Q3:** Why was 0.7 chosen to map rerank scores to binary judgments? Please report sensitivity to (or a continuous calibration curve).
> >
> > **A6:** We thank the reviewer for the question regarding the selection of the 0.7 threshold.
> >
> > This threshold is used during the training phase to convert continuous scores from our reranker (`bge-reranker`) into binary 'good'/'bad' judgment labels for training our agent. To determine a suitable value for our setup, we conducted a sensitivity analysis on the test set. The results, showing final answer accuracy (%) versus the threshold, are presented below:
> >
> > |Threshold|0.40|0.45|0.50|0.55|0.60|0.65|0.68|**0.70 (Ours)**|0.72|0.75|0.80|
> > |-|-|-|-|-|-|-|-|-|-|-|-|
> > |EM|28.5|29.1|29.8|30.5|30.9|30.8|31.0|**31.2**|31.2|31.1|30.1|
> >
> > The analysis shows that performance peaks at **31.2%** with a threshold of 0.70. Furthermore, we observe that performance is relatively stable in the 0.65-0.75 range, indicating that our training process is not overly sensitive to minor variations around this optimal point. Based on this empirical evidence, we selected 0.7 as a robust and justified choice for our specific reranker. **We have added this analysis to the Sec.4.3 (Table 6).**
> >
> > **W5:** Notation/definition issues. Eq. (1) contains a formatting error. In Eq. (2), the history term is referenced but not explicitly defined earlier.
> >
> > **A7:** We thank the reviewer for spotting these notation details. We have corrected them in the revised paper:
> >
> >   1. **Eq. (1) Formatting:**
> > We appreciate the reviewer pointing out the formatting issue. We have corrected the equation in the revised paper to ensure mathematical precision. The revised objective function is:
> > equation in the revised paper to ensure mathematical precision. The revised objective function is: $$\max_{\pi_\theta} \mathbb{E}{x \sim \mathcal{D}, y \sim \pi\theta(\cdot|x) }[R(x, y)] - \beta D_{KL} [\pi_\theta(y | x) || \pi_{\text{ref}}(y | x)]$$
> >
> >   2. **Definition of $\tau_{t-1}$ in Eq. (2):**
> > We have added an explicit definition in the revised paper for $\tau_{t-1}$ in the text immediately preceding Eq. (2). It represents the **interaction trajectory**—the sequence of all past actions and observations up to step $t-1$.

---

> > > ### Author Response · Authors · 2025-11-21
> > > **Responses to the Reviewer k1mq [3/4]**
> > >
> > > **Q4.1:** How is the fidelity of reasoning structures preserved during paraphrasing?
> > >
> > > **A8:** We thank the reviewer for this question. We ensure the model adheres to the prescribed reasoning structure using a two-part mechanism:
> > >
> > > 1. **Instructional Prompt:** As detailed in Section 3.4, a comprehensive system prompt defines the valid actions (`SEARCH`, `JUDGE`, `ANSWER`) and their required syntax (e.g., using `<search>` tags).
> > > 2. **Automated Format Enforcement:** If the model generates an output in an invalid format, our system provides a specific corrective prompt and requires the model to retry. The prompt explicitly reminds it of the correct format: *"My previous action is invalid. If I want to search, I should put the query between \<search> and \</search>. If I want to give the final answer, I should put the answer between \<answer> and \</answer>. Let me try again."*
> > >
> > > We observed that while these guided retries occur in early training, their frequency quickly diminishes as the model learns to internalize the required structure. This combination of instructional prompting and format enforcement ensures high fidelity to the reasoning format.
> > >
> > > **Q4.2:** Which metrics (e.g., semantic equivalence, reasoning graph preservation) and annotation protocols validate that the paraphrases keep the same reasoning demands?
> > >
> > > **A9:** We thank the reviewer for this question. We ensure the model adheres to the prescribed reasoning structure using a two-part mechanism: an **instructional prompt** and an **automated format enforcement** loop.
> > >
> > > To provide quantitative evidence for the effectiveness of this approach, we tracked the number of invalid actions that required a guided retry during a full training run. At each training step, a total of 2048 actions are generated (512 batch size × 4 turns). The figure in this url https://f.anonymous-uploads.xyz/uploads/curve_structure.png plots the number of these actions that were invalid.
> > >
> > > As the figure shows, the number of invalid actions peaks at the beginning of training (approx. 50) but decreases sharply within the first 80 steps. Subsequently, the number of errors stabilizes at a near-zero level for the remainder of the training. This provides strong empirical evidence that the model is not merely being forced into a format but is effectively learning and internalizing the desired reasoning structure. **We have added this analyzis to the Appendix A.7.**
> > >
> > > **Q5.1:** The abstract states a decomposition into “correctness” and “utility,” yet Sec. 3.3 seems to present a binary R_judge based on a rerank threshold (Eq. (3)).
> > >
> > > **A10:** We thank the reviewer for this question, which has helped us clarify our reward structure. We have **revised Section 3.3 to provide a clearer description**. Our framework decomposes the reward into two distinct components, as stated in the abstract:
> > >
> > > 1. **Correctness Reward ($R_{\text{answer}}$):** This is a standard, rule-based reward for the final `ANSWER` action, based on Exact Match (EM). The reward is binary: 1 for a correct answer and 0 otherwise.
> > > 2. **Utility Reward ($R_{\text{judge}}$):** As described in Sec. 3.3 (Eq. (3)), $R_{\text{judge}}$ evaluates the agent's `JUDGE` action. The reward is determined by whether the agent's judgment ($j_t$) matches the ground-truth label ($j^*_t$). The $R\_{\text{match}}$ provides a reward of 0.3 for both types of correct judgments: correctly identifying useful information ($j_t=j^\*_t=\text{Yes}$) and correctly discarding useless information ($j_t=j^\*_t=\text{No}$). The $R\_{\text{mismatch}}$ implements asymmetric penalties: incorrectly accepting useless information incurs a large penalty of -0.6, while incorrectly discarding useful information incurs a smaller penalty of -0.3.
> > >
> > > **Q5.2:** Please write the complete training reward,
> > >
> > > **A11:** The complete training reward for a full trajectory $\tau = (s_0, a_0, ..., s_T, a_T)$ is the discounted sum of rewards obtained at each step:
> > >
> > > $$R(\tau) = \sum_{t=1}^{T} \gamma^{t-1} r_t$$
> > >
> > > Here, $\gamma$ is the discount factor, and the per-step reward $r_t$ combines a final **correctness** reward with intermediate **utility** rewards, depending on the action taken:
> > > $$
> > > r_t = \mathbb{I}(a_t = Judge) \cdot R_{\text{judge}} + \mathbb{I}(a_t = Answer) \cdot R_{\text{answer}}
> > > $$
> > >
> > > This structure provides both a clear terminal objective ($R_{\text{answer}}$) and dense, step-by-step guidance ($R_{\text{judge}}$) to effectively train the agent.
> > >
> > > **Q5.3:** and include ablations that zero out each term to quantify its contribution.
> > >
> > > **A12:**  As demonstrated in the ablation study **(Answer 5, A5)**, the inclusion of the `JUDGE` action improves performance by **+0.8** (29.6 vs. 28.8), quantifying the value of adaptive search termination. Furthermore, incorporating the dense utility reward ($R_{\text{judge}}$) provides an additional **+1.6** gain, confirming the effectiveness of our composite reward structure.

---

> ### Author Response · Authors · 2025-11-21
> **Responses to the Reviewer k1mq [4/4]**
>
> **Q6.1:** Since ReSeek relies on multi-turn interaction plus an auxiliary JUDGE step, please report latency per question, tokens per question, and training/inference FLOPs/compute.
>
> **A13:** We report the computational costs of ReSeek, which relies on multi-turn interactions and an auxiliary JUDGE step.
>
> - **Latency per Question:** On a single NVIDIA H20 GPU, the total inference latency for an average of $T=4$ interaction turns is measured at **2.55 seconds** per question.
> - **Tokens per Question:** The token overhead is modest. A 4-turn interaction adds approximately **80-200 tokens** to the input context compared to a non-retrieval baseline. This primarily stems from appending retrieved evidence (20-50 tokens per turn) for the REASON step. The output of the auxiliary JUDGE step is negligible, adding only about 3 tokens per turn.
> - **Training and Inference Compute:**
>     - **Training:** The model was fine-tuned on 8 NVIDIA H20 GPUs for **24 hours**. Based on the H20's peak BF16/FP16 performance (296 TFLOPs) and an estimated 95% hardware utilization, the total computational budget for a single training run is calculated as:
>     $$ \text{Total FLOPs} \approx 8 \text{ GPUs} \times 24 \text{ hr} \times 3600 \frac{\text{s}}{\text{hr}} \times 296 \frac{\text{TFLOPs}}{\text{GPU}} \times 95\% \text{ Utilization} \approx \textbf{194.3 PetaFLOPs} $$
>     - **Inference:** Evaluating the entire test set on the same 8-GPU setup takes approximately **2 hours**, corresponding to a total compute of **16.19 PetaFLOPs**.
>
> **Q6.2:** A turn-budget vs. latency trade-off plot would clarify whether improvements reflect gains rather than redundancy.
>
> **A14:** Thank you for the suggestion. We have performed the requested analysis and added the following plot (https://f.anonymous-uploads.xyz/uploads/performance_vs_latency.png), which visualizes the trade-off between performance and latency.
>
> As the plot shows, latency (blue line) increases linearly with each turn, representing a predictable cost. In contrast, accuracy (orange line) shows substantial gains up to **T=4** but plateaus immediately after.
>
> This result confirms that our multi-turn approach provides genuine gains, not redundancy. Our choice of T=4 captures the majority of the performance benefits at an effective computational cost, striking an optimal balance. **We have added this analyzis to the Appendix A.8.**

---

> ### Author Response · Authors · 2025-11-26
>
> May we kindly inquire if the provided responses have adequately addressed any questions you might have had? If there remains a requirement for further explanations or clarifications? We wish to express our sincere gratitude for your meticulous evaluation and for generously investing a significant amount of your time in reviewing our paper. Your feedback would be greatly valued.

---

### Author Response · Authors · 2025-12-01
**Global Response**

We sincerely thank all PCs, SACs, ACs, and Reviewers for their time and efforts when handling our paper. We are grateful for the constructive feedback, which has significantly improved the quality and rigor of our work.

All reviewers appreciate the contributions of our method and the importance of the problem:

*   **Reviewers k1mq, uaiX, and P3XZ** recognized that our proposed **JUDGE action** and **self-correction mechanism** effectively address the limitations of sparse rewards in search agents, enabling dynamic path adaptation and error recovery.
*   **Reviewer k1mq** highlighted that our approach instills "a form of meta-cognitive control without heavy backtracking."
*   **Reviewers uaiX and sAzq** commended the introduction of the **FictionalHot benchmark**, noting that it helps mitigate data contamination and allows for a more comprehensive evaluation of reasoning capabilities.

As suggested by the reviewers, we have revised the manuscript as follows:

*   **Enhanced Experimental Rigor:** We added statistical significance tests (p-values) and reported mean/standard deviation across multiple runs to ensure result stability (Table 1). We also included detailed ablations on reward components, turn budgets, and embedding models (Sec 4.3 & Appendix).
*   **Clarification of Methodology:** We revised section 3.2 to clarify the "soft removal" mechanism in Eq. (2) and section 3.3 to explicitly define the reward decomposition ($R_{ans}$ and $R_{judge}$), ensuring strict alignment between the paper and our code implementation.
*   **Reranker and Threshold Details:** We added a detailed explanation of the reranker-based silver label generation (Sec 3.3) and a sensitivity analysis justifying the selection of the 0.7 threshold (Sec 4.3 & Table 6).
*   **Dataset Details:** We provided comprehensive statistics, generation pipelines, and a human evaluation for the FictionalHot dataset to validate its quality and consistency (Sec 4.1 & Appendix A.2).
*   **Efficiency Analysis:** We added an analysis of latency, token usage, and computational costs to demonstrate the trade-off between performance and inference budget (Appendix A.8).

We have highlighted all modifications by underlining them in the revised manuscript.

Beyond these manuscript revisions, **we summarize below how we have addressed the specific concerns of each reviewer**:

*   **For Reviewers k1mq and uaiX**, who primarily focused on methodological clarity and experimental sufficiency, we have fully incorporated their suggestions into the revised paper and provided all requested additional results to ensure completeness.
*   **Regarding Reviewer P3XZ**'s concerns about code implementation, we sincerely appreciate the effort spent scrutinizing our codebase. However, we respectfully note that there were some misunderstandings regarding specific implementation details. We have provided a point-by-point clarification in our individual response to resolve these misconceptions and demonstrate the correctness of our code.
*   **Addressing Reviewer sAzq**'s inquiries about *FictionalHot* and conceptual novelty, we have expanded the appendix with comprehensive dataset statistics and validation. Furthermore, we respectfully argue that ReSeek represents a significant step forward **as the first work to explicitly apply information verification within the Search Agent domain**. Our extensive experiments further validate that this verification mechanism is an effective strategy for enhancing agent reliability.

---

### Note · Authors · 2025-12-08

I have read and agree with the venue's withdrawal policy on behalf of myself and my co-authors.